# Exact recovery and Bregman hard clustering of node-attributed Stochastic Block Model

**Maximilien Dreveton**
School of Computer and Communication Sciences
EPFL, Lausanne, Switzerland
`maximilien.dreveton@epfl.ch`

**Felipe S. Fernandes**
Systems Engineering and Computer Science
Federal University of Rio de Janeiro
Rio de Janeiro, Brazil
`felipesc@cos.ufrj.br`

**Daniel R. Figueiredo**
Systems Engineering and Computer Science
Federal University of Rio de Janeiro
Rio de Janeiro, Brazil
`daniel@cos.ufrj.br`

## Abstract

Network clustering tackles the problem of identifying sets of nodes (communities) that have similar connection patterns. However, in many scenarios, nodes also have attributes that are correlated with the clustering structure. Thus, network information (edges) and node information (attributes) can be jointly leveraged to design high-performance clustering algorithms. Under a general model for the network and node attributes, this work establishes an information-theoretic criterion for the exact recovery of community labels and characterizes a phase transition determined by the Chernoff-Hellinger divergence of the model. The criterion shows how network and attribute information can be exchanged in order to have exact recovery (e.g., more reliable network information requires less reliable attribute information). This work also presents an iterative clustering algorithm that maximizes the joint likelihood, assuming that the probability distribution of network interactions and node attributes belong to exponential families. This covers a broad range of possible interactions (e.g., edges with weights) and attributes (e.g., non-Gaussian models), as well as sparse networks, while also exploring the connection between exponential families and Bregman divergences. Extensive numerical experiments using synthetic data indicate that the proposed algorithm outperforms classic algorithms that leverage only network or only attribute information as well as state-of-the-art algorithms that also leverage both sources of information. The contributions of this work provide insights into the fundamental limits and practical techniques for inferring community labels on node-attributed networks.

## 1   Introduction

Community detection or network clustering–the task of identifying sets of similar nodes in a network–is a fundamental problem in network analysis [4, 1, 18], with applications in diverse fields such as digital humanities, data science and biology. In the classic formulation, a set of communities must be determined from the connection patterns among the nodes of a single network. A simple random graph model with community structure, the Stochastic Block Model (SBM), has been the canonical model to characterise theoretical limitations and evaluate different community detection algorithms [1].

37th Conference on Neural Information Processing Systems (NeurIPS 2023).

However, nodes of many real-world networks have attributes or features that can reveal their identity as an individual or within a group. For example, the age, gender and ethnicity of individuals in a social network [27], the title, keywords and co-authors of papers in a citation network [32], or the longitude and latitude of meteorological stations in weather forecast networks [8]. In some scenarios, such attributes can be leveraged alone to identify node communities (clusters) without even using the network.

Thus, a modern formulation for community detection must consider network information (edges) and node information (attributes). Indeed, recent works have designed community detection algorithms that can effectively leverage both sources of information to improve performance. Various methods have been proposed for clustering node-attributed networks, including modularity optimisation [14], belief propagation [15], expected-maximisation algorithms [24, 34], information flow compression [33], semidefinite programming [37], spectral algorithms [2, 7, 25] and iterative likelihood based methods [8].

A fundamental problem in this new formulation is fusing both sources of information: how important is network information in comparison to node information given a problem instance? Intuitively, this depends on the noise associated with network edges and node attributes. For example, if edges are reliable then the clustering algorithm should prioritize them when determining the communities. However, most prior approaches adopt some form of heuristic when merging the two sources of information [14, 16, 37]. A rigorous approach to this problem requires a mathematical model, and one has been recently proposed.

The *Contextual Stochastic Block Model* (CSBM) is a generalization of the SBM where each node has a random attribute that depends on its community label. While the model formulation is general (in terms of distribution for edges and attributes), CSBM has only been rigorously studied in the restrictive setting where the pairwise interactions are binary (edges are present or not) and the node attributes are Gaussian [2, 8, 15]. In this scenario, the phase transitions for exactly recovering the community labels and for detecting them better than a random guess has been established. Moreover, the comparison with the respective phase transitions in SBM [1] and in Gaussian mixture model [11, 26] (with no network) highlight the value of jointly leveraging network and node information in recovering community labels.

However, real networks often depart from binary edges and Gaussian attributes. Indeed, in many scenarios network edges have weights that reveal information about that interaction and nodes have discrete or non-Gaussian attributes. This work tackles this scenario by considering a CSBM where edges have weights and nodes have attributes that follow some arbitrary distributions. Under this general model, this work is the first to characterise the phase transition for the exact recovery of community labels. In particular, the *Chernoff-Hellinger divergence*, initially defined just for binary networks [3], is extended to this more general model. This divergence effectively captures the difficulty of distinguishing different communities and thus plays a crucial role in determining the limits of exact recovery. The analysis reveals an additional term in the divergence that quantifies the information provided by the attributes of the nodes. Moreover, it quantifies the trade-off between network and node information in meeting the threshold for exact recovery.

The CSBM generates weighted networks that are complete (all possible edges are present) when edge weights follow a continuous distribution. However, most weighted real networks are sparse. To model sparse weighted networks and to provide a practical community detection algorithm, we consider a CSBM whose weights belong to *zero-inflated distributions*. More precisely, we suppose that conditioned on observing an edge, the distribution of the weight of this edge belongs to an exponential family. Similarly, the node attribute distributions are also assumed to belong to an exponential family. Working with exponential families is motivated by two factors. Firstly, exponential families encompass a broad range of parametric distributions, including the commonly used Bernoulli, Poisson, Gaussian, or Gamma distributions. Secondly, there exists an intricate connection between exponential families of distributions and Bregman divergences, which has proven to be a powerful tool for developing algorithms across a variety of problems such as clustering, classification, and dimensionality reduction [6, 13].

This connection between Bregman divergences and exponential families has been previously explored in the context of clustering dense networks (all possible edges are present) [23]. In contrast, this work proposes an iterative algorithm that maximizes the log-likelihood of the model, for both dense and sparse networks. This is a key difference with many previous works which either study only dense

weighted networks [9, 24] or binary networks with Gaussian attributes [2, 15, 34]. Simulations on synthetic networks demonstrate that our algorithm outperforms state-of-the-art approaches in various settings, providing practical techniques for achieving accurate clustering results.

The article is structured as follows. The relevant related work is discussed in Section 2. Section 3 introduces the model under consideration along with the main theoretical contributions on exact recovery. Section 4 focuses on sparse networks with edge weights and node attributes drawn from exponential families and introduces an iterative algorithm for clustering such networks. Numerical results and comparisons to prior works are presented in Section 5, and Section 6 concludes the paper.

**Notations** Let $\mathrm{Ber}(p)$ denote a Bernoulli random variable (r.v.) with parameter $p$, $\mathrm{Nor}(m, \sigma)$ a Gaussian r.v. with mean $m$ and standard deviation $\sigma$, and $\mathrm{Exp}(\lambda)$ an exponential r.v. with mean $\lambda^{-1}$. The notation $[K]$ refers to the set $\{1, \cdots, K\}$, while $A_i$ stands for the $i$-th row of matrix $A$.

## 2 Related work

### 2.1 Exact recovery in SBM with edge weights and node attributes

Community detection in classic SBM (binary edges) is a well-understood problem with strong theoretical results concerning exact recovery and efficient algorithms with guaranteed accuracy [1, 39]. However, extending the classic SBM to weighted networks (non-binary edges) with arbitrary distributions is an ongoing research area. Most existing work in this scenario has been restricted to the *homogeneous model*[1], where edge weights within and across communities are determined by two respective distributions. Moreover, existing works often restrict to categorical or real-valued weights [22, 36], or to multiplex networks (multiple edge types) with independent and identically distributed layers [29]. However, a recent work has provided a strong theoretical foundation the homogeneous model with arbitrary distributions [5], highlighting the role of the Rényi divergence as the key information-theoretic quantity for the homogeneous model.

In non-homogeneous models, a more complex divergence called the Chernoff-Hellinger divergence is the appropriate information-theoretic quantity for exact community recovery [3]. However, the expression of the Chernoff-Hellinger divergence as originally defined in [3] for binary networks does not have an intuitive interpretation, and its extension to non-binary (weighted) networks is challenging. For example, the exact recovery threshold for non-homogeneous SBM whose edges are categorical random variables has been established [38], but this threshold is expressed as a condition involving the minimization of a mixture of Kullback-Leibler divergences over the space of probability distributions. Although the relationship between Kullback-Leibler and Chernoff divergences are known (see for example [35, Theorem 30-32]), the specific technical lemma required to link them to the Chernoff-Hellinger divergence is not straightforward (see [38, Claim 4]).

Another generalization of the SBM allows for nodes to have attributes that provide information about their community, such as the Contextual SBM (CSBM) [15]. The CSBM has only been rigorously studied in the setting where edges are binary and node attributes follow a Gaussian distribution. In this scenario, the phase transition for exact recovery for the community labels has been established [2, 8, 15]. A natural generalization is to investigate the model where network edges have weights and nodes have attributes that follow arbitrary distributions. Indeed, this is one of the main contributions of this work: Expression (3.4) gives a straightforward yet crucial formula for the phase transition for exact recovery, also providing a natural interpretation for the influence of both the network and node attributes. Moreover, Expression (3.4) also applies when no node attribute is available, thus providing the exact recovery threshold for a non-homogeneous model and arbitrary edge weight distribution, a significant advancement in the state of the art.

### 2.2 Algorithms for clustering weighted networks with node attributes

Algorithms leveraging different approaches have been proposed to tackle community detection in networks with edge weights and node attributes. A common principled approach is to determine the community assignment that maximizes the likelihood function of a model for the data. However, optimizing the likelihood function is computationally intractable even for binary networks. Thus,

---

[1]Also known as the *planted partition model.*

approximation schemes such as variational inference and pseudo-likelihood methods are often adopted. For instance, [24] introduced a variational-EM algorithm for clustering non-homogeneous weighted SBM with arbitrary distributions. Another approach for clustering node-attributed SBM whose edge weights and attribute distributions belong to exponential families is [23]. These two approaches assume that the network is dense (all edges are present and have non-zero edge weight). However, most real networks are very sparse (most node pairs do not have an edge) and this work focuses on this scenario. Another very recent work tackling sparse networks is the *IR_sLs* algorithm from [8], although its theoretical guarantees assume binary networks with Gaussian attributes.

The iterative clustering algorithm presented in this work maximizes the pseudo-likelihood likelihood by assuming that the probability distribution of network edges and node attributes belong to exponential families. This yields a direct connection with Bregman divergences and establishes an elegant expression for the likelihood function. This connection has also been leveraged in [23], however, their model is restricted to dense weighted networks (all edges are present). This work (more specifically, Lemma 2), demonstrates that this connection can also be applied to sparse weighted networks (using zero-inflated distributions to model the weights). This extension enhances the applicability of pseudo-likelihood algorithms using Bregman divergence to a broader class of scenarios, namely weighted sparse networks with node attributes.

## 3 Model and exact recovery in node-attributed SBM

### 3.1 Model definition

Consider a population of $n$ objects, called *nodes*, partitioned into $K \geq 2$ disjoint sets, called *blocks* or communities. A node-labelling vector $z = (z_1, \cdots, z_n) \in [K]^n$ represents this partitioning so that $z_i$ indicates the block of node $i$. The labels (blocks) of nodes are random variables assumed to be independent and identically distributed such that $\mathbb{P}(z_i = k) = \pi_k$ for some vector $\pi \in (0,1)^K$ verifying that $\sum_k \pi_k = 1$. The nodes interact in unordered pairs giving rise to undirected edges, and $\mathcal{X}$ is the measurable space of all possible pairwise interactions. Additionally, each node has an attribute that is an element of a measurable space $\mathcal{Y}$. Let $X \in \mathcal{X}^{N \times N}$ denote the symmetric matrix such that $X_{ij}$ represents the interaction between node pair $(ij)$, and by $Y = (Y_1, \cdots, Y_n) \in \mathcal{Y}^n$ the node attribute vector.

Assume that interactions and attributes are independent conditionally on the community labels of the nodes. Let $f_{k\ell}(x)$ denote the probability that two nodes in blocks $k$ and $\ell$ have an interaction $x \in \mathcal{X}$, and $h_k(y)$ denote the probability that a node in block $k \in [K]$ has an attribute $y \in \mathcal{Y}$. Thus,

$$\mathbb{P}(X, Y \mid z) = \prod_{1 \leq i < j \leq n} f_{z_i z_j}(X_{ij}) \prod_{i=1}^{n} h_{z_i}(Y_i). \tag{3.1}$$

In the following, the interaction spaces $\mathcal{X}, \mathcal{Y}$ might depend on $n$, as well as the respective interaction probabilities $f, h$. The number of nodes $n$ will increase to infinity while $K$ and $\pi$ are constant. For an estimator $\hat{z} \in [K]^n$ of $z$, we define the *classification error* as

$$\text{loss}(z, \hat{z}) = \min_{\tau \in \mathcal{S}_K} \text{Ham}(z, \tau \circ \hat{z}),$$

where $\mathcal{S}_K$ is the set of permutations of $[K]$ and $\text{Ham}(\cdot, \cdot)$ is the hamming distance between two vectors. An estimator $\hat{z} = \hat{z}(X, Y)$ achieves *exact recovery* if $\mathbb{P}(\text{loss}(z, \hat{z}) \geq 1) = o(1)$.

### 3.2 Exact recovery threshold in node-attributed SBM

The difficulty of classifying empirical data in one of $K$ possible classes is traditionally measured by the *Chernoff information* [12]. More precisely, in the context of network clustering, let $\text{CH}(a, b) = \text{CH}(a, b, \pi, f, h)$ denote the hardness of distinguishing nodes that belong to block $a$ from block $b$. This quantity is defined by

$$\text{CH}(a, b) = \sup_{t \in (0,1)} \text{CH}_t(a, b), \tag{3.2}$$

where

$$\text{CH}_t(a, b) = (1 - t) \left[ \sum_{c=1}^{K} \pi_c \, \text{D}_t(f_{bc} \| f_{ac}) + \frac{1}{n} \text{D}_t(h_b \| h_a) \right] \tag{3.3}$$

is the *Chernoff coefficient* of order $t$ across blocks $a$ and $b$, and $D_t(f\|g) = \frac{1}{t-1}\log\int f^t(x)g^{1-t}(x)dx$ is the *Rényi divergence* of order $t$ between two probability densities $f, g$ [35]. The key quantity assessing the possibility or impossibility of exact recovery in SBM is then the minimal Chernoff information across all pairs of clusters. We denote it by $I = I(\pi, f, h)$, and it is defined by

$$I = \min_{\substack{a,b\in[K]\\a\neq b}} \mathrm{CH}(a,b). \tag{3.4}$$

The following Theorem provides the information-theoretic threshold for exact recovery in node-attributed SBM.

**Theorem 1.** *Consider model* (3.1) *with $\pi_a > 0$ for all $a \in [K]$. Denote by $a^*, b^*$ the two hardest blocks to estimate, that is $\mathrm{CH}(a^*, b^*) = I$. Suppose that $t \in (0,1) \mapsto \lim_{n\to\infty} \frac{n}{\log n}\mathrm{CH}_t(a^*, b^*)$ exists and is strictly concave. Then the following holds:*

*(i) exact recovery is information-theoretically impossible if $\lim_{n\to\infty}\frac{n}{\log n}I < 1$;*

*(ii) exact recovery is information-theoretically possible if $\lim_{n\to\infty}\frac{n}{\log n}I > 1$.*

The proof for Theorem 1 is provided in the supplemental material. The main ingredient of the proof is the asymptotic study of log-likelihood ratios. More precisely, let $L(z) = \mathbb{P}(X, Y \mid z)$. The application of Chernoff bounds provides an upper bound on $-\log\mathbb{P}(\log L(z) \geq \log L(z'))$, where $z$ denotes the correct block structure and $z' \in [K]^n$ is another node-labelling vector (see Lemma 1 in the supplement). We can use this upper bond to prove that the maximum likelihood estimator (MLE) achieves exact recovery if $I > n^{-1}\log n$. Reciprocally, if $I < n^{-1}\log n$, we show that whp there exist some "bad" nodes $i$ for which $z_i \neq \arg\max_{a\in[k]}\mathbb{P}(X, Y \mid z_{-i}, z_i = a)$. In other words, even in a setting where an oracle would reveal $z_{-i}$ (*i.e.,* the correct block assignment of all nodes except node $i$), the MLE would fail at recovering $z_i$. Establishing this fact requires to lower bound $-\log\mathbb{P}(\log L(z) \geq \log L(z'))$ where $z' \in [K]^n$ is such that $\mathrm{Ham}(z, z') = 1$ (*i.e.,* $z'$ correctly labels all nodes except one). This lower bound is obtained using large deviation results for general random variables [10, 30]. To apply these results, the strict concavity of the limit $n(\log n)^{-1}I$ is needed. In most practical settings, this assumption is verified, except in some edge cases (see Examples 1 and 2). Let us now provide some examples of applications of Theorem 1.

**Example 1** (Binary SBM with no attributes). *Suppose that $f_{ab} \sim \mathrm{Ber}(\alpha_{ab}n^{-1}\log n)$ where $\alpha_{ab}$ are constants. A Taylor-expansion of the Rényi divergence between Bernoulli distributions leads to*

$$I = (1 + o(1))\frac{\log n}{n}\min_{a\neq b}\sup_{t\in(0,1)}\left(\sum_c \pi_c\left[t\alpha_{bc} + (1-t)\alpha_{ac} - \alpha_{bc}^t\alpha_{ac}^{1-t}\right]\right),$$

*which indeed coincides with the expression of the Chernoff-Hellinger divergence defined in [3]. We also note that the limit $n(\log n)^{-1}I$ is strictly concave as long as the $\alpha_{ab}$ are not all equals[2].*

**Example 2** (Binary SBM with Gaussian attributes). *Suppose that $f_{ab} \sim \mathrm{Ber}(\alpha_{ab}n^{-1}\log n)$ and $h_a \sim \mathrm{Nor}(\mu_a\log n, \sigma^2 I_d)$, where $\alpha_{ab}$ and $\mu_a$ are independent of $n$. Then,*

$$I = (1 + o(1))\frac{\log n}{n}\min_{a\neq b}\sup_{t\in(0,1)}\left(\sum_c \pi_c\left[t\alpha_{bc} + (1-t)\alpha_{ac} - \alpha_{bc}^t\alpha_{ac}^{1-t}\right] + t\frac{\|\mu_b - \mu_a\|_2^2}{2\sigma^2}\right).$$

*In particular, the technical conditions of Theorem 1 are verified if we rule out the uninformative case where all the $\alpha_{ab}$'s and the $\mu_a$'s are equal to each other. Thus, exact recovery is possible if*

$$\min_{a\neq b}\sup_{t\in(0,1)}\left(\sum_c \pi_c\left[t\alpha_{bc} + (1-t)\alpha_{ac} - \alpha_{bc}^t\alpha_{ac}^{1-t}\right] + t\frac{\|\mu_b - \mu_a\|_2^2}{2\sigma^2}\right) > 1.$$

*Further assuming that $\alpha_{ab} = \alpha 1(a = b) + \beta 1(a \neq b)$ (homogeneous interactions) and $\pi = \left(\frac{1}{K}, \cdots, \frac{1}{K}\right)$ (uniform block probabilities), the expression of $I$ simplifies to*

$$I = (1 + o(1))\frac{\log n}{n}\left[\frac{\left(\sqrt{\alpha} - \sqrt{\beta}\right)^2}{K} + \frac{\Delta^2}{8\sigma^2}\right],$$

---

[2]Indeed, since the matrix $\alpha = (\alpha_{ab})$ is symmetric, this implies that there exists $a \neq b$ such that $\exists c^* \in [k]: \alpha_{ac^*} \neq \alpha_{bc^*}$. The function $f(t) = \sum_c \pi_c\left[t\alpha_{bc} + (1-t)\alpha_{ac} - \alpha_{bc}^t\alpha_{ac}^{1-t}\right]$ is continuous and strictly concave, hence $\sup_{t\in(0,1)}f(t)$ is strictly concave.

*where $\Delta = \min\limits_{a \neq b} \|\mu_a - \mu_b\|_2$. This last scenario recovers the recently established threshold for exact recovery in the Contextual SBM [8].*

**Example 3** (Semi-supervised clustering in SBM)**.** *Consider binary interactions given by*

$$f_{ab} \sim \begin{cases} \mathrm{Ber}(\alpha n^{-1} \log n) & \text{if } a = b, \\ \mathrm{Ber}(\beta n^{-1} \log n) & \text{otherwise,} \end{cases}$$

*where $\alpha, \beta$ are independent of $n$. Consider a semi-supervised model in which the vector of attributes $Y$ is a noisy oracle of the true community labels $z$. More precisely, for a node $i$ such that $z_i = k$, let*

$$h_k(y) = \mathbb{P}(Y_i = y) = \begin{cases} 1 - \eta & \text{if } y = 0, \\ \eta_1 & \text{if } y = k, \\ \frac{\eta_0}{K-1} & \text{if } y \in [K] \backslash \{k\}, \end{cases}$$

*with $\eta_0 + \eta_1 = \eta$. A bit of algebra shows that exact recovery is possible if*

$$\left( \sqrt{\alpha} - \sqrt{\beta} \right)^2 - \frac{2}{\log n} \log \left( 1 - \eta + 2(K-1)^{-1/2} \sqrt{\eta_0 \eta_1} \right) > K.$$

*When $\eta_0 = 0$ (perfect oracle), the condition simplifies to $(\sqrt{a} - \sqrt{b})^2 - \frac{2\log(1-\eta)}{\log n} > K$. Note that the oracle term is non-negligible only if $-\log(1-\eta) \gtrsim \log n$, as previously established [31]. This last condition is very strong since it implies $\eta \gtrsim 1 - 1/n$, and hence the oracle must provide the correct label for almost all nodes.*

# 4 Bregman hard clustering of sparse weighted node-attributed networks

In this section, we will propose an algorithm for clustering sparse, weighted networks with node attributes. When present, the weights are sampled from an exponential family, and the node attributes also belong to an exponential family. In Section 4.1, we provide some reminder of exponential families. We derive the likelihood of the model in Section 4.2, and present the algorithm in Section 4.3.

## 4.1 Exponential family

An exponential family $\mathcal{E}_\psi$ is a parametric class of probability distributions whose densities can be canonically written as $p_{\theta,\psi}(x) = e^{<\theta,x> - \psi(\theta)}$, where the density is taken with respect to an appropriate measure, $\theta \in \Theta$ is a function of the parameters of the distribution that must belong to an open convex space $\Theta$, and $\psi$ is a convex function.

We consider the model defined in (3.1), such that $f_{ab}$ are *zero-inflated distributions* and are given by

$$f_{ab}(x) = (1 - p_{ab})\delta_0(x) + p_{ab}\tilde{f}_{ab}(x), \tag{4.1}$$

where $p_{ab} \in [0,1]$ is the interaction probability between blocks $a$ and $b$, $\delta_0(x)$ is the Dirac delta at zero, and $\tilde{f}_{ab}$ is a probability density with no mass at zero. Note that this model can represent sparse weighted networks, as edges between nodes in blocks $a$ and $b$ are absent with probability $1 - p_{ab}$.

Finally, suppose that the distributions $\{\tilde{f}_{ab}\}$ and $\{h_a\}$ belong to exponential families. More precisely,

$$\tilde{f}_{ab}(x) = e^{<\theta_{ab},x> - \psi(\theta_{ab})} \quad \text{and} \quad h_a(y) = e^{<\eta_a,y> - \phi(\eta_a)}, \tag{4.2}$$

for some parameters $\theta_{ab}, \eta_a$ and functions $\psi, \phi$. The following lemma provides the expression of the Chernoff divergence of this model.

**Lemma 1.** *Let $f_{ab}$ and $h_a$ be defined as in (4.1)-(4.2). Suppose that $p_{ab} = \alpha_{ab}\delta$ where $\alpha_{ab}$ is constant and $\delta \ll 1$. We have*

$$I = (1 + o(1)) \min_{a \neq b} \sup_{t \in (0,1)} \left\{ \sum_{c \in [K]} \pi_c \left[ t p_{ac} + (1-t) p_{bc} - p_{ac}^t p_{bc}^{1-t} e^{-J_\psi(\theta_{ac} \| \theta_{bc})} \right] + J_\phi(\eta_a \| \eta_b) \right\},$$

*where $J_\psi(\theta_{ab} \| \theta_{bc}) = t\psi(\theta_{ac}) + (1-t)\psi(\theta_{bc}) - \psi(t\theta_{ac} + (1-t)\theta_{bc})$.*

In combination with Theorem 1, Lemma 1 provides the expression for the exact recovery threshold for node-attributed networks with interaction and attribute distributions given by (4.1)-(4.2).

## 4.2 Log-likelihood

Given a convex function $\psi$, the *Bregman divergence* $d_\psi \colon \mathbb{R}^m \times \mathbb{R}^m \to \mathbb{R}_+$ is defined by

$$d_\psi(x, y) \;=\; \psi(x) - \psi(y) - <x - y, \nabla\psi(y)> .$$

The log-likelihood of the density $p_{\psi,\theta}$ of an exponential family distribution is linked to the Bregman divergence by the following relationship (see for example [6, Equation 13])

$$\log p_{\psi,\theta}(x) \;=\; -d_{\psi^*}(x, \mu) + \psi^*(x), \tag{4.3}$$

where $\mu = \mathbb{E}_{p_{\psi,\theta}}(X)$ is the mean of the distribution, and $\psi^*$ denotes the *Legendre transform* of $\psi$, defined by $\psi^*(t) \;=\; \sup_\theta\{<\theta, t> -\psi(\theta)\}$. The following Lemma provides an expression for the log-likelihood of $f_{ab}$ when $f_{ab}$ is a distribution belonging to a zero-inflated exponential family.

**Lemma 2.** *Let $f_{ab}$ be a probability density as defined in (4.1)-(4.2). For $x, y \in (0, 1)$, let $d_{\mathrm{KL}}(x, y)$ be the Kullback-Leibler divergences between $\mathrm{Ber}(x)$ and $\mathrm{Ber}(y)$, and let $H(x) = x \log x + (1 - x) \log(1 - x)$, with the usual convention $0 \log 0 = 1$. Then,*

$$-\log f_{ab}(x) \;=\; d_{\mathrm{KL}}(x\|p_{ab}) + s d_{\psi^*}(x, \mu_{ab}) - s\psi^*(x) - H(s),$$

*where $s = 1(x \neq 0)$.*

*Proof.* To express $\log f_{ab}(x)$, we first note that

$$\log f_{ab}(x) \;=\; (1 - s)\log(1 - p_{ab}) + s \log p_{ab} + s \log(\tilde{f}_{ab}(x)),$$

where $c = 1(x \neq 0)$. The result follows by adding and subtracting $H(b)$ in the previous expression and expressing $\log \tilde{f}_{ab}(x)$ with a Bregman divergence as in (4.3). $\square$

Suppose that $X, Y$ follow the model (3.1) with probability distributions given by (4.1)-(4.2). Let $A$ be a binary matrix such that $A_{ij} = 1(X_{ij} \neq 0)$. We have

$$-\log \mathbb{P}(X, Y \mid z) \;=\; \sum_i \left\{ \frac{1}{2} \sum_{j \neq i} \left[ d_{\mathrm{KL}}(A_{ij}, p_{z_i z_j}) + A_{ij} d_{\psi^*}\left(X_{ij}, \mu_{z_i z_j}\right) \right] + d_{\phi^*}(Y_i, \nu_{z_i}) \right\} + c,$$

where the additional term $c$ is a function of $X, Y$ but does not depend on $z$. Denoting $Z \in \{0, 1\}^{n \times K}$ the one-hot membership matrix such that $Z_{ik} = 1(z_i = k)$, observe that $p_{z_i z_j} = \left(Z p Z^T\right)_{ij}$ where $p$ is a symmetric matrix with the interaction probabilities between different blocks, $\mu_{z_i z_j} = \left(Z \mu Z^T\right)_{ij}$ where $\mu$ is a symmetric matrix with the expected value of the interaction between different blocks (edge weights), and $\nu_{z_i} = \left(Z^T \nu\right)_i$ where $\nu$ is a vector with the expected value of the attribute for different blocks. Thus, up to some additional constants, the negative log-likelihood $-\log \mathbb{P}(X, Y \mid Z)$ is equal to

$$\sum_i \left\{ \frac{1}{2} d_{\mathrm{KL}}\left(A_{i\cdot}, \left(Z p Z^T\right)_{i\cdot}\right) + \frac{1}{2} d'_{\psi^*}\left(X_{i\cdot}, \left(Z \mu Z^T\right)_{i\cdot}\right) + d_{\phi^*}\left(Y_i, \left(Z^T \nu\right)_i\right) \right\} + c, \tag{4.4}$$

where $d'_{\psi^*}(B, C) = \sum_{j=1}^n 1(B_j \neq 0) d_{\psi^*}(B_j, C_j)$ for two vectors $B, C \in \mathbb{R}^n$.

## 4.3 Clustering by iterative likelihood maximisation

Following the log-likelihood expression derived in (4.4), we propose an iterative clustering algorithm that places each node in the block maximising $\mathbb{P}(X, Y \mid z_{-i}, z_i = a)$ for $1 \leq a \leq K$, the likelihood that node $i$ is in community $a$ given the community labels of the other nodes, $z_{-i}$. Let $Z^{(ia)}$ denote the membership matrix obtained from $Z$ by placing node $i$ in block $a$, and let $L_{ia}(Z^{(ia)})$ denote the contribution of node $i$ to the negative log-likelihood when node $i$ is placed in block $a$. Equation (4.4) shows that

$$L_{ia}(Z) \;=\; \frac{1}{2} d_{\mathrm{KL}}\left(A_{i\cdot}, \left(Z p Z^T\right)_{i\cdot}\right) + \frac{1}{2} d'_{\psi^*}\left(X_{i\cdot}, \left(Z \mu Z^T\right)_{i\cdot}\right) + d_{\phi^*}\left(Y_i, \left(Z^T \nu\right)_i\right), \tag{4.5}$$

where the $p$, $\mu$ and $\nu$ in the equation above must be estimated from $X$, $Y$, and the community membership matrix $Z$. Let $\hat{p} = \hat{p}(A, Z)$, $\hat{\mu} = \hat{\mu}(X, Z)$, and $\hat{\nu} = \hat{\nu}(Y, Z)$ denote the estimators for $p$, $\mu$ and $\nu$, respectively. Their values can be computed as follows:

$$
\begin{aligned}
\hat{p}(A, Z) &= \left(Z^T Z\right)^{-1} Z^T A Z \left(Z^T Z\right)^{-1}, \\
\hat{\mu}(X, Z) &= \left(Z^T A Z\right)^{-1} Z^T X Z, \\
\hat{\nu}(Y, Z) &= \left(Z^T Z\right)^{-1} Z^T Y.
\end{aligned}
\tag{4.6}
$$

Note that the matrix inverse $\left(Z^T Z\right)^{-1}$ can be easily computed since $Z^T Z$ is a $K$-by-$K$ diagonal matrix. This approach is described in Algorithm 1.

---

**Algorithm 1:** Bregman hard clustering for node-attributed SBM.

---

**Input:** Interactions $X \in \mathcal{X}^{n \times n}$, attributes $Y \in \mathcal{Y}^n$, convex functions $\psi^*, \phi^*$, clustering $Z_0$

1 Let $Z = Z_0$
2 **repeat**
3     Compute $\hat{p}, \hat{\mu}, \hat{\nu}$ according to (4.6)
4     Let $Z^{\text{new}} = 0_{n \times K}$
5     **for** $i = 1, \ldots, n$ **do**
6        Let $Z^{(ia)}$ be the membership matrix obtained from $Z$ by placing node $i$ in community $a$
7        Find $k^* = \arg\max_{a \in [K]} L_{ia}\left(Z^{(ia)}\right)$, where $L_{ia}\left(Z^{(ia)}\right)$ is defined in (4.5);
8        Let $Z^{\text{new}}_{ik} = 1(k = k^*)$ for all $k = 1, \ldots, K$
9     Let $Z = Z^{\text{new}}$
10 **until** *convergence*;
**Return:** Node-membership matrix $Z$

---

A fundamental aspect of many likelihood maximization iterative algorithms such as Algorithm 1 is the initial membership assignment, $Z_0$. This initial assignment often has a profound influence on the final membership assignment, and thus, it is important to have an adequate initialization. In the numerical section, we proceed as follows. We construct the matrix $W \in \mathbb{R}^{n \times 2K}$ such that the first $K$ columns of $W$ are the first $K$ eigenvectors of the graph normalised Laplacian, while the last $K$ columns of $W$ are the first $K$ eigenvectors of the Gram matrix $Y Y^T$.

## 5 Numerical experiments

### 5.1 Performance of Algorithm 1

We first compare in Figure 1 the performance of Algorithm 1 in terms of exact recovery (fraction of times the algorithm correctly recovers the community of *all* nodes) with the theoretical threshold for exact recovery proved in the paper (red curve in the plots) in two settings: Figure 1a shows binary weight with Gaussian attributes, and Figure 1b shows zero-inflated Gaussian weights with Gaussian attributes. A solid black (resp., white) square means that over 50 trials, the algorithms failed 50 times (resp., succeeded 50 times) at exactly recovering the block structure.

### 5.2 Comparison with other algorithms

In this section, we compare Algorithm 1 with other algorithms presented in the literature. We used the Adjusted Rand Index (ARI) [21] between the predicted clusters and the ground truth ones to evaluate the performance of each algorithm.

In Figure 2, we compare Algorithm 1 with the variational-EM algorithm of [24] and the algorithm of [23] (which is also based on Bregman divergences, but tailored for dense networks). Because both of these algorithms are designed for dense networks, we observe that Algorithm 1 has overall better performance on sparse networks.

We also compare Algorithm 1 with the *IR_sLs* algorithm from [8]. This is one of the most recent algorithms for node-attributed SBM and it comes with theoretical guarantees (for binary networks

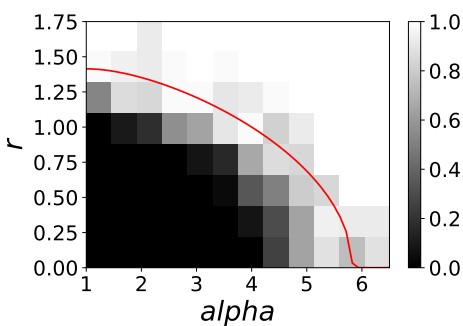

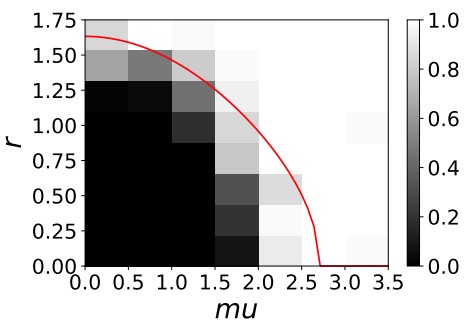

(a) Binary weights with Gaussian attributes

(b) zero-inflated Gaussian weights with Gaussian attributes.

Figure 1: Phase transition of exact recovery. Each pixel represents the empirical probability that Algorithm 1 succeeds at exactly recovering the clusters (over 50 runs), and the red curve shows the theoretical threshold.

(a) $n = 500$, $K = 2$, $f_{\text{in}} = \text{Ber}(\alpha n^{-1} \log n)$, $f_{\text{out}} = \text{Ber}(n^{-1} \log n)$. The attributes are 2d-spherical Gaussian with radius $(\pm r \sqrt{\log n}, 0)$ and identity covariance matrix.

(b) $n = 600$, $K = 3$, $f_{\text{in}} = (1 - \rho)\delta_0 + \rho \text{Nor}(\mu, 1)$, $f_{\text{out}} = (1 - \rho)\delta_0 + \rho \text{Nor}(0, 1)$ with $\rho = 5n^{-1} \log n$. The attributes are 2d-spherical Gaussian whose means are the vertices of a regular polygon on the circle of radius $r \sqrt{\log n}$.

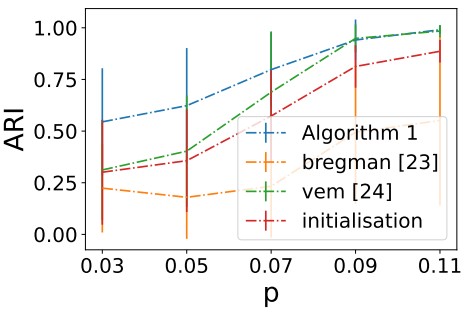

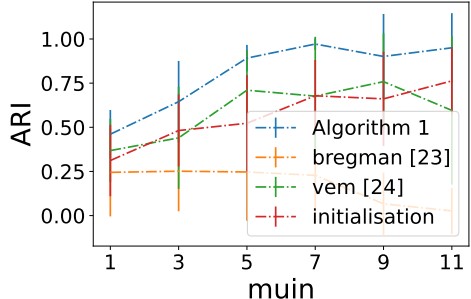

Figure 2: Comparison of Algorithm 1 with algorithms of [23] and [24]. Error bars show the standard deviations over 25 realisations. Attributes are 2d-spherical Gaussian attributes with radius $(\pm 1, 0)$.

(a) $n = 100$, $K = 2$, $f_{\text{in}} = (1 - p_{\text{in}})\delta_0 + p_{\text{in}} \text{Poi}(5)$, $f_{\text{out}} = (1 - 0.03)\delta_0 + 0.03 \text{Poi}(1)$.

(b) $n = 100$, $K = 2$, $f_{\text{in}} = (1 - 0.07)\delta_0 + 0.07 \text{Poi}(\mu_{\text{in}})$, $f_{\text{out}} = (1 - 0.04)\delta_0 + 0.04 \text{Poi}(1)$.

with Gaussian attributes). We also compare with the EM algorithm of [34], *attSBM*, which is designed for binary networks with Gaussian attributes. Finally, we compare with baseline methods for clustering using the network or the attributes alone. *EM-GMM* refers to fitting a Gaussian Mixture Model via EM on attribute data $Y$, and *sc* refers to *spectral clustering* on network data $X$.

Figure 3 shows the results for binary networks with Gaussian attributes. Algorithm 1 successfully learns from both the signal coming from the network and the attributes, even in scenarios where one of them is non-informative. Moreover, Algorithm 1 has better performance than the two other node-attributed clustering algorithms, and those algorithms also show a large variance[3]. We also note that *IR_sLs* and *attSBM* are both tailor-made for binary edges and Gaussian attributes. Even in such a setting, Algorithm 1 outperforms these two algorithms. We show in the supplement material that when the network is weighted and the attributes non-Gaussian, *IR_sLs* and *attSBM* perform poorly.

### 5.3 Evaluation using real datasets

The following three benchmark datasets were used to evaluate and compare the proposed algorithm: *CiteSeer* ($n = 3279$, $m = 9104$, $K = 6$, $d = 3703$), *Cora* ($n = 2708$, $m = 10556$, $K = 7$,

---

[3] As the large variance makes the figures less readable, we provide all results in the supplemental material.

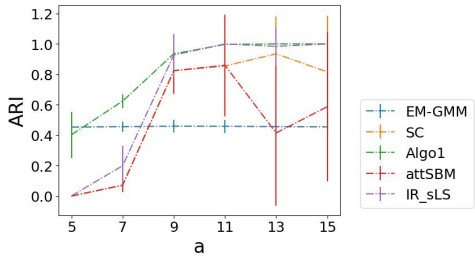
(a) Varying edge distribution.

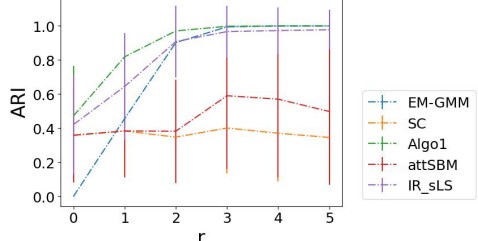
(b) Varying attribute distributions.

Figure 3: Performance on binary networks with Gaussian attributes. We take $n = 600$, $K = 2$, $f_{\text{in}} = \text{Ber}(an^{-1} \log n)$, $f_{\text{out}} = \text{Ber}(5n^{-1} \log n)$, and 2-dimensional Gaussian attributes with unit variance and mean $(\pm r, 0)$. Results are averaged over 60 runs.

$d = 1433$), and *Cornell* ($n = 183$, $m = 298$, $K = 5$, $d = 1703$) (all available in Pytorch Geometric). For each network, the original node attribute vector was reduced to have dimension $d = 10$ by selecting the 10 best features according to the chi-square test. Algorithm 1 assumed a multivariate Gaussian distribution with $d = 10$ for node attributes and Bernoulli edges (these networks have no edge weights). The initialization for Algorithm 1 and attSBM used spectral clustering of both the node similarity matrix (using node attributes) and network edges.

| Dataset | CiteSeer | Cora | Cornell |
|---------|----------|------|---------|
| Algorithm 1 | **0.20** | **0.12** | **0.49** |
| *attSBM* | 0.17 | 0.09 | 0.46 |
| *EM-GMM* | 0.13 | 0.06 | 0.37 |
| *sc* | 0.00 | 0.00 | 0.02 |

Table 1: Average ARI results (over independent runs) for the three benchmark datasets.

Table 1 shows that Algorithm 1 outperformed the other three algorithms. Spectral clustering (*sc*) has near zero performance, indicating that the network structure in these data sets has little information concerning the clusters of the nodes. Moreover, both Algorithm 1 and attSBM (that leverage network and node attributes) outperform EM-GMM that use only node attributes. These preliminary results indicate that Algorithm 1 is promising even in real data sets with little pre-processing.

## 6 Conclusion

This work made the following contributions to community detection in node-attributed networks: i) extended the known thresholds for exact recovery in binary SBM to non-binary (weighted) networks with node attributes, providing a clean expression for a new information-theoretic quantity, known in the binary setting as the Chernoff-Hellinger divergence; ii) proposed an iterative algorithm based on the likelihood function that can infer community memberships from a problem instance. The algorithm leverages the framework of Bregman divergences and is simple and computationally efficient. Numerical experiments indicate the superiority of this algorithm when compared to recent state-of-the-art approaches.

## Acknowledgements

The first author would like to thank Lasse Leskelä for helpful discussions and comments.

This work has been partially funded by the Brazilian-Swiss Joint Research Program (grant IZBRZ2_186313), the Brazilian National Council for Scientific and Technological Development (CNPq), and the Carlos Chagas Filho Research Foundation of the State of Rio de Janeiro (FAPERJ).

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

# A Establishing the exact recovery threshold

The proof of Theorem 1 is structured as follows. We start by establishing some concentration results on the block sizes in Section A.1. In Section A.2, we delve into the asymptotic analysis of log-likelihood ratios, establishing fundamental results. Building upon these findings, we prove the converse statement, demonstrating the impossibility of exact recovery below the threshold, in Section A.3. Conversely, in Section A.4, we establish the positive statement, demonstrating the possibility of exact recovery above the threshold. To complement the proof, Section A.5 presents a set of technical combinatorial lemmas.

**Notations** In the following, we denote by $z \in [K]^n$ the true block-labelling vector, and by $z' \in [K]^n$ another block-labelling vector, and the conditional probabilities are denoted by $\mathbb{P}_z(\cdot) = \mathbb{P}(\cdot \mid z)$.

## A.1 Preliminaries on the block sizes

For any $z' \in [K]^n$, we for all $c \in [K]$ define $\hat{\pi}_c(z')$, the empirical size of block $c$, as

$$\hat{\pi}_c(z') = \frac{|\{i \in [n] \colon z_i' = c\}|}{n}.$$

For any $0 < \xi < 1$, we define

$$\mathcal{Z}_\xi = \{z \in [K]^n \colon (1-\xi)\pi_c \leq \hat{\pi}_c(z) \leq (1+\xi)\pi_c \quad \forall c \in [K]\}. \tag{A.1}$$

Recalling that the true block-labelling vector $z$ is sampled from $\pi^{\otimes n}$, we have by concentration of multinomial distributions (see for example [5, Lemma A1])

$$\mathbb{P}(\mathcal{Z}_\xi) \geq 1 - 2K \exp\left(-\frac{\xi^2}{3}n \min_{c \in [K]} \pi_c\right).$$

In the following, we chose $\xi = \frac{\log\log n}{\sqrt{n}}$, and hence $\mathbb{P}(\mathcal{Z}_\xi) = 1 - o(1)$.

## A.2 Asymptotic study of log-likelihood ratios

This section studies the asymptotic behaviour of $\mathbb{P}_z(L(z) \geq L(z'))$, where $L(z') = \mathbb{P}_{z'}(X, Y)$ the likelihood of the block-labelling vector $z'$ given the observed data $X, Y$.

### A.2.1 Upper-bound

The following lemma provides an upper bound on $\mathbb{P}_z(L(z) \geq L(z'))$.

**Lemma 3.** *Let $z, z' \in \mathcal{Z}$ be two block labelling vectors such that $\mathrm{Ham}(z, z') = m \geq 1$, and let $L(z) = \mathbb{P}(X, Y \mid z)$ be the likelihood of labelling $z$. We have*

$$\mathbb{P}_z(L(z') \geq L(z)) \leq e^{-(1+o(1))\left(1 - \frac{m}{n\pi_{\min}}\right)mnI}.$$

*Moreover, let $d = \sup_{t \in (0,1)} \min_{\substack{a \neq b \in [K] \\ c \in [K]}} (1-t)\, \mathrm{D}_t(f_{ac}\|f_{bc})$. Then we also have*

$$\mathbb{P}_z(L(z') \geq L(z)) \leq e^{-m(n-m)d}.$$

*Proof.* Using Chernoff's bound, we have for all $t \geq 0$,

$$\mathbb{P}_z\left(\log\frac{\mathbb{P}_{z'}(X,Y)}{\mathbb{P}_z(X,Y)} > \epsilon\right) \leq e^{-t\epsilon}\, \mathbb{E}_z\left[e^{t \log \frac{\mathbb{P}_{z'}(X,Y)}{\mathbb{P}_z(X,Y)}}\right] = e^{-t\epsilon - (1-t)\,\mathrm{D}_t(P_z\|P_{z'})}, \tag{A.2}$$

where the probability measure $\mathbb{P}_z$ is defined on $\mathcal{X} \times \mathcal{Y}$ by (3.1) in the main text. The linearity of the Rényi divergence with respect to product distributions implies

$$\mathrm{D}_t(P_z\|P_{z'}) = \frac{1}{2}\sum_{i \neq j} \mathrm{D}_t\left(f_{z_i z_j}\|f_{z_i' z_j'}\right) + \sum_{1 \leq i \leq n} \mathrm{D}_t(h_{z_i}\|h_{z_i'})$$

$$= \frac{1}{2}\sum_{1 \leq a,b,c,d \leq K} M_{(ab)(cd)}\, \mathrm{D}_t(f_{ac}\|f_{bd}) + \sum_{1 \leq a,b \leq K} N_{ab}\, \mathrm{D}_t(h_a\|h_b), \tag{A.3}$$

where
$$N_{ab} = \left|\{i \in [n]\colon (z_i, z_i') = (a, b)\}\right|,$$
$$M_{(ab)(cd)} = \left|\{i \neq j\colon (z_i, z_i') = (a, b) \text{ and } (z_j, z_j') = (c, d)\}\right|.$$

Moreover, Lemma 6 ensures that
$$M_{(ab)(cd)} = N_{ab}\left(N_{cd} - 1_{(ab)=(cd)}\right) = N_{cd}\left(N_{ab} - 1_{(ab)=(cd)}\right).$$

Therefore,

$$\sum_{1 \leq a,b,c,d \leq K} M_{(ab)(cd)}\, \mathrm{D}_t(f_{ac}\|f_{bd}) \geq \sum_{a \neq b}\sum_c M_{(ab)(cc)}\, \mathrm{D}_t(f_{ac}\|f_{bc}) + \sum_a \sum_{c \neq d} M_{(aa)(cd)}\, \mathrm{D}_t(f_{ac}\|f_{ad})$$

$$= 2\sum_{a \neq b} N_{ab} \sum_c N_{cc}\, \mathrm{D}_t(f_{ac}\|f_{bc}). \tag{A.4}$$

Using $N_{cc} = \left|z^{-1}(c)\right| - \sum_{d \neq c} N_{cd}$ and that $\sum_c \sum_{d \neq c} N_{cd} = m$, we obtain that

$$\frac{1}{2} \sum_{1 \leq a,b,c,d \leq K} M_{(ab)(cd)}\, \mathrm{D}_t(f_{ac}\|f_{bd}) \geq n \sum_{a \neq b} N_{ab} \sum_c \hat{\pi}_c(z)\left(1 - \frac{\sum_{d \neq c} N_{cd}}{n\hat{\pi}_c(z)}\right) \mathrm{D}_t(f_{ac}\|f_{bc})$$

$$\geq n\left(1 - \frac{m}{n\min_c \hat{\pi}_c(z)}\right) \sum_{a \neq b} N_{ab} \sum_c \hat{\pi}_c(z)\, \mathrm{D}_t(f_{ac}\|f_{bc}).$$

Hence,

$$(1-t)\,\mathrm{D}_t\left(P_{z'}\|P_z\right) \geq n\left(1 - \frac{m}{n\min_c \hat{\pi}_c(z)}\right) \sum_{a \neq b} N_{ab}(1-t)\left(\sum_c \hat{\pi}_c(z)\, \mathrm{D}_t(f_{ac}\|f_{bc}) + \frac{1}{n}\mathrm{D}_t(h_a\|h_b)\right)$$

$$= n\left(1 - \frac{m}{n\min_c \hat{\pi}_c(z)}\right) \sum_{a \neq b} N_{ab}\mathrm{CH}_t(a, b, \hat{\pi}(z)).$$

Using the concentration of $\hat{z}$, we have $\hat{\pi}_c(z) = (1 + o(1))\pi_c$ for all $c$, and hence

$$\sup_{t \in (0,1)} (1-t)\,\mathrm{D}_t\left(P_{z'}\|P_z\right) \geq (1 + o(1))\left(1 - \frac{m}{n\pi_{\min}}\right) mnI,$$

where we used $\sum_{a \neq b} N_{ab} = m$ and $I \leq \mathrm{CH}(a, b, \hat{\pi}(\hat{z}))$. The first upper bound on $\mathbb{P}_z(L(z') \geq L(z))$ follows by taking $\epsilon = 0$ and the supremum over $t \in (0, 1)$ in (A.2).

Finally, let $d = \sup_{t \in (0,1)} \min_{\substack{a \neq b \in [K] \\ c \in [K]}} (1-t)\,\mathrm{D}_t(f_{ac}\|f_{bc})$. Combining (A.3) and (A.4) leads to

$$\sup_{t \in (0,1)} (1-t)\,\mathrm{D}_t(P_z\|P_{z'}) \geq d\sum_{a \neq b} N_{ab} \sum_c N_{cc} = dm(n - m),$$

and this ends the proof. $\qquad\square$

### A.2.2 Lower bound

We now focus on lower-bounding $\mathbb{P}_z\left(L(z) \geq L(z')\right)$ when $\mathrm{Ham}(z, z') = 1$. In particular, the condition $\mathrm{Ham}(z, z') = 1$ implies that there exists a unique node $u \in [n]$ such that $z_u \neq z_u'$. Let $a, b \in [K]$ such that $z_u = a$ and $z_u' = b$. By definition of the likelihood, we have $L(z) - L(z') = \Delta_{ub}(X, Y, z)$ where

$$\Delta_{ub}(X, Y, z) = \sum_{\substack{v=1 \\ v \neq u}}^n \log \frac{f_{bz_v}}{f_{az_v}}(X_{uv}) + \log \frac{h_b}{h_a}(Y_u). \tag{A.5}$$

Before studying the large deviation rates of likelihood ratios such as $\mathbb{P}_z\left(\Delta_{ub} > 0\right)$ and $\mathbb{P}_z\left(\Delta_{ub} > 0, \Delta_{vb} > 0\right)$, we first recall some large deviation result for arbitrary sequences of random variables [30, 10].

**Proposition 1** (Strong large deviations for arbitrary sequences of random variables – Theorem 3.3 of [10])**.** *Let $T_n$ be a sequence of random variables whose moment generating function $\phi_n(t) = \mathbb{E}\left[e^{tT_n}\right]$ is non-vanishing and analytic in the region $\Omega = \{z \in \mathbb{C}\colon |z| < a\}$ for some $a > 0$. Let $k_n$ be a sequence of real numbers, and let $\psi_n(t) = k_n^{-1}\log\phi_n(t)$. Let $(m_n)$ be a bounded sequence of real numbers such that there exists a sequence $(\tau_n)$ verifying $\psi_n'(\tau_n) = m_n$ and $0 < \tau_n < \alpha_0 < \alpha$. Suppose that:*

1. *there exists $b > 0$ such that $|\psi_n(z)| \leq b$ for all $z \in \Omega$;*

2. *there exists $a > 0$ such that $\psi_n''(\tau_n) > c$;*

3. *there exists $\delta_0 > 0$ such that $\sup_{\delta<|t|<\lambda\tau_n}\left|\frac{\phi_n(\tau_n+it)}{\phi_n(\tau_n)}\right| = o(k_n^{-1/2})$ for any given $\delta, \lambda$ such that $0 < \delta < \delta_0 < \lambda$.*

*Then,*

$$\mathbb{P}\left(T_n > a_n m_n\right) \;=\; \frac{1+o(1)}{\tau_n\sqrt{2\pi a_n \psi''(\tau_n)}}e^{-a_n(m_n\tau_n - \psi_n(\tau_n))}.$$

Proposition 1 is an extension of Cramer's large deviation theorem for sums of iid r.v. to sequences of arbitrary random variables (see [10, Remark 3.6]). We will apply it to the study of $\mathbb{P}\left(\Delta_{ub} > 0\right)$.

**Lemma 4.** *Let $z \in \mathcal{Z}_\xi$ (where $\mathcal{Z}_\xi$ is defined in (A.1)) with $\xi = o(1)$. Let $u \in [n]$ such that $z_u = a$ and let $\Delta_{ub}$ be given in (A.5). Suppose that there exists $\delta_n = \omega(1)$ such that $\beta\colon t \in (0,1) \mapsto -\lim_{n\to\infty} n\delta_n^{-1}\mathrm{CH}_t(a,b)$ exists and is strictly convex. Then, for any sequence $\epsilon_n$ such that $\delta_n\epsilon_n \to \epsilon \in \{\beta'(t), t \in (0,1)\}$ we have*

$$\mathbb{P}_z\left(\Delta_{ub} > \delta_n\epsilon_n\right) \;\sim\; \frac{1}{t_\epsilon\sqrt{2\pi\delta_n\beta''(t_\epsilon)}}e^{-\delta_n\beta^*(t_\epsilon)},$$

*where $\beta^*(x) = \sup_{t>0}\{tx - \beta(t)\}$ is the [Legendre transform](#) of $\beta$, and $t_\epsilon$ is such that $\beta'(t_\epsilon) = \epsilon$. In particular $0 \in \{\beta'(t), t \in (0,1)\}$, and therefore for any sequence $\epsilon_n = o(\delta_n^{-1})$ we have*

$$\mathbb{P}_z\left(\Delta_{ub} > \delta_n\epsilon_n\right) \;\sim\; \frac{1}{t_0\sqrt{2\pi\delta_n\beta''(t_0)}}e^{-n\mathrm{CH}(a,b)}.$$

*Proof of Lemma 4.* We will apply Proposition 1 with $\Omega = \{z \in \mathbb{C}\colon |z| < 1\}$ and $k_n = \delta_n^{-1}$ to obtain the stated results.

Let us first compute $\phi_n(t) = \mathbb{E}_z\left[e^{t\Delta_{ub}}\right]$ and $\psi_n(t) = \delta_n^{-1}\log\phi_n(t)$ for $t \in (0,1)$. We first notice that for all $v \in [n]$,

$$\mathbb{E}_z\left[e^{t\log\frac{f_{bz_v}}{f_{az_v}}(X_{uv})}\right] \;=\; \mathbb{E}\left[\left(\frac{f_{bz_v}}{f_{az_v}}\right)^t(X_{uv})\right] \;=\; e^{-(1-t)\,\mathrm{D}_t(f_{bz_v}\|f_{az_v})},$$

by definition of the Rényi divergence. Similar computations show that

$$\mathbb{E}_z\left[e^{t\log\frac{h_b}{h_a}(Y_u)}\right] \;=\; e^{-(1-t)\,\mathrm{D}_t(h_b\|h_a)}.$$

Therefore,

$$\phi_n(t) \;=\; e^{t\log\frac{\pi_b}{\pi_a}-n\mathrm{CH}_t(a,b,z)},$$

where

$$\mathrm{CH}_t(a,b,z) \;=\; (1-t)\left[\sum_{c=1}^K \hat{\pi}_c\,\mathrm{D}_t\left(f_{bc}\|f_{ac}\right) + \frac{1}{n}\,\mathrm{D}_t\left(h_b\|h_a\right)\right], \tag{A.6}$$

with $\hat{\pi}_c = n^{-1}\left|z^{-1}(k)\right|$. We note that since $z \in \mathcal{Z}_\xi$ we have $\mathrm{CH}_t(a,b,z) = (1 + O(\xi))\mathrm{CH}_t(a,b)$. Thus, $\psi_n(t) = -(1 + o(1))n\mathrm{CH}_t(a,b)$. Moreover, the assumption $\mathrm{CH}(a,b) = \Theta\left(n^{-1}\delta_n\right)$ implies

that $\mathrm{CH}_t(a,b) = \Theta\left(n^{-1}\delta_n\right)$ for all $t \in (0,1)$ since the Rényi divergences of orders $t \in (0,1)$ are equivalent [35, Theorem 16]. Hence, $\delta_n^{-1}\psi_n(t) = (1+o(1))\beta(t)$. Since $\beta$ is well-defined and strictly convex on $(0,1)$, this ensures that Assumptions 1 and 2 of Proposition 1 are verified. Moreover, we notice that $e^{\beta(t)}$ is the m.g.f of some r.v. $X$, and let $\varphi(t) = \mathbb{E}e^{itX}$ be the characteristic function of $X$. Then, $\frac{\phi_n(t^*+it)}{\phi_n(t^*)} = \left(\frac{\varphi(t^*+it)}{\varphi(t^*)}\right)^{\delta_n}$. Since $\frac{\varphi(t^*+it)}{\varphi(t^*)}$ is the characteristic function of some r.v., its module is strictly less than 1 on an interval not containing 0, and hence Assumption 3 of Proposition 1 is verified.

Finally, Lemma 8 ensures that $\beta'(0) \leq 0$ and $\beta'(1) \geq 0$ and hence $0 \in \{\beta'(t), t \in (0,1)\}$. $\qquad\square$

### A.2.3 Asymptotic independence

Finally, the following lemma shows that the events $\Delta_{ub} > 0$ and $\Delta_{vb} > 0$ are asymptotically independent.

**Lemma 5.** *Let $z \in \mathcal{Z}_\xi$, $u, v \in z^{-1}(a)$, and $b \in [K]\backslash\{a\}$. Then,*

$$\frac{\mathbb{P}_z\left(\Delta_{ub} > 0, \Delta_{vb} > 0\right)}{\mathbb{P}_z\left(\Delta_{ub} > 0\right)\mathbb{P}_z\left(\Delta_{vb} > 0\right)} = 1 + o(1).$$

*Proof.* Let $\xi(X_{uv}) = \log\frac{f_{aa}}{f_{ab}}(X_{uv})$. We have

$$\Delta_{ub} = \log\frac{\pi_b}{\pi_a} + \sum_{\substack{w=1 \\ v\notin\{u,v\}}}^{n} \log\frac{f_{bz_w}}{f_{az_w}}(X_{uw}) + \log\frac{h_b}{h_a}(Y_u) + \log\frac{f_{ba}}{f_{aa}}(X_{uv}),$$

$$\Delta_{vb} = \log\frac{\pi_b}{\pi_a} + \sum_{\substack{w=1 \\ v\notin\{u,v\}}}^{n} \log\frac{f_{bz_w}}{f_{az_w}}(X_{vw}) + \log\frac{h_b}{h_a}(Y_v) + \log\frac{f_{ba}}{f_{aa}}(X_{uv}).$$

Therefore,

$$\Delta_{ub} = U_u - \xi(X_{uv}) \quad \text{and} \quad \Delta_{vb} = U_v - \xi(X_{uv})$$

where $U_u$ and $U_v$ are iid. Therefore,

$$\mathbb{P}_z\left(\Delta_{ub} > 0, \Delta_{vb} > 0\right) - \mathbb{P}_z\left(\Delta_{ub} > 0\right)\mathbb{P}_z\left(\Delta_{vb} > 0\right) = \mathrm{Cov}_z(1(\Delta_{ub} > 0), 1(\Delta_{vb} > 0)).$$

Conditioning on $U_u$ and $U_v$, we observe that

$$\mathrm{Cov}_z(1(\Delta_{ub} > 0), 1(\Delta_{vb} > 0)) = \int\int \mathrm{Cov}_z\left(\phi_u(\xi), \phi_{u'}(\xi)\right)\mu(du)\mu(du'),$$

where $\phi_t(\xi) = 1(t > \xi)$ and $\mu = \mathrm{Law}(\xi)$. Because $\phi_u$ and $\phi_{u'}$ are monotonous, the Fortuin–Kasteleyn–Ginibre (FKG) inequality [20, 17] (see also [19, Section 2.2]) implies that $\mathrm{Cov}\left(\phi_u(\xi), \phi_{u'}(\xi)\right) \geq 0$. Therefore,

$$\frac{\mathbb{P}_z\left(\Delta_{ub} > 0, \Delta_{vb} > 0\right)}{\mathbb{P}_z\left(\Delta_{ub} > 0\right)\mathbb{P}_z\left(\Delta_{vb} > 0\right)} \geq 1.$$

Let us now derive an upper bound for this ratio. First, Lemma 4 with $\epsilon_n = 0$ and $\delta_n = \log n$ implies

$$\mathbb{P}\left(\Delta_{ub} > 0\right) = \frac{1 + o(1)}{t^*\sqrt{2\pi\log n\beta''(t^*)}}e^{-n\mathrm{CH}(a,b)},$$

and the same relation holds for $\mathbb{P}\left(\Delta_{vb} > 0\right)$. Moreover, conditionally on $X_{uv}$ we have

$$\mathbb{P}_z\left(\Delta_{ub} > 0, \Delta_{vb} > 0\right) = \int_{x\in\mathcal{X}} \left(\gamma\left(x\right)\right)^2 f_{aa}(x)dx,$$

where $\gamma(x) = \mathbb{P}_z\left(U > \xi(x)\right)$. Since $\mathrm{CH}_t(a,b) = o(1)$, then $f_{ac}$ and $f_{bc}$ are mutually contiguous[4] for all $c$ (see [35, Theorem 25]). In particular, this implies that $\xi(x) < \infty$ for all $x \in \mathcal{X}$ such that

---

[4]Let $(p_n)$ and $(q_n)$ be two sequences of distributions. Then $p_n$ is contiguous with respect to $q_n$ if for all sequence of event $\mathcal{A}_n$ such that $q_n(\mathcal{A}_n) = o(1)$ we also have $p_n(\mathcal{A}_n) = o(1)$.

$f_{aa}(x) > 0$. Thus, we can apply Lemma 4 with $\epsilon_n = \xi(x)$ and $\delta_n = \log n$, verifying $\delta_n \epsilon_n \to 0$ to obtain

$$\gamma(x) = \frac{1 + o(1)}{t^* \sqrt{2\pi \log n \beta''(t^*)}} e^{-n\text{CH}(a,b)},$$

and this ends the proof.

$\square$

## A.3 Impossibility of exact recovery

For $a, b \in [K]$ two block indexes, we recall the quantity $\text{CH}(a, b)$ defined in (3.2) of the main text denotes the hardness to distinguish a node in block $a$ from a node in block $b$. We suppose that $I = \min_{a \neq b} \text{CH}(a, b) = \Theta\left(\frac{\log n}{n}\right)$ with $\lim_{n \to \infty} \frac{n}{\log n} I < 1$. We prove the failure of the MAP estimator for exact recovery in three steps:

(i) We start by showing that we can restrict the study to the node labelling vector $z$ for which the relative size of the communities are close to their expectations, *i.e.*, $\frac{|z^{-1}(a)|}{n} = (1 + o(1))\pi_a$ for all $a \in [K]$;

(ii) Let $M_{ab}(z)$ be the number of nodes in block $a$ for which changing their community label to block $b$ results in an increase of likelihood. We show that $\mathbb{E}M_{ab}(z) \gg 1$ when $a, b$ are the two hardest blocks to distinguish.

(iii) We show that $M_{ab}(z) > 0$ almost surely using the second-moment method.

**(i) Conditioning on well-behaving community sizes.** Recall the definition of $\mathcal{Z}_\xi$ in (A.1). We established in Section A.1 that $\mathbb{P}(\mathcal{Z}_\xi) = 1 - o(1)$ for $\xi = \frac{\log \log n}{\sqrt{n}}$. Hence,

$$
\begin{aligned}
\mathbb{P}(\text{MAP fails}) &= \sum_{z \in \mathcal{Z}} \mathbb{P}_z\left(z^{\text{MAP}} \neq z\right) \mathbb{P}(z) \\
&\geq \sum_{z \in \mathcal{Z}_\xi} \mathbb{P}_z\left(z^{\text{MAP}} \neq z\right) \mathbb{P}(z) \\
&\geq \mathbb{P}(\mathcal{Z}_\xi) \inf_{z \in \mathcal{Z}_\xi} \mathbb{P}_z\left(z^{\text{MAP}} \neq z\right) \\
&\geq (1 - o(1)) \inf_{z \in \mathcal{Z}_\xi} \mathbb{P}_z\left(z^{\text{MAP}} \neq z\right).
\end{aligned}
$$

The rest of the proof is devoted to show that $\mathbb{P}_z\left(z^{\text{MAP}} \neq z\right) = 1 - o(1)$ for all $z \in \mathcal{Z}_\xi$.

**(ii) Expected number of bad nodes.** Given a block structure $z \in \mathcal{Z}_\xi$, an arbitrary node $u \in [n]$ such that $z_u = a$ and a block $b \neq a$. We define by $\tilde{z}$ the block structure obtained from $z$ by swapping the block of node $u$ to $b$, *i.e.*,

$$\forall v \in [n]: \tilde{z}_v = (1 - \delta_{vu})z_v + \delta_{vu}b.$$

Suppose that $X, Y$ are generating from a true block structure $z$, and let

$$\Delta_{ub}(X, Y, z) = \log \mathbb{P}(\tilde{z} \mid X, Y) - \log \mathbb{P}(z \mid X, Y) \tag{A.7}$$

the change in the MAP estimation of $z$ obtained by swapping the label of node $u$ from its true block $a$ to a wrong block $b$. By the definition of $\tilde{z}$, we have using Bayes' law

$$
\begin{aligned}
\Delta_{ub}(X, Y, z) &= \log\left(\frac{\mathbb{P}(z_u = b \mid X, Y, z_{-u})}{\mathbb{P}(z_u = a \mid X, Y, z_{-u})}\right) \\
&= \log\left(\frac{\pi_b \mathbb{P}(X, Y \mid z_{-u}, z_u = b)}{\pi_a \mathbb{P}(X, Y \mid z_{-u}, z_u = a)}\right).
\end{aligned}
$$

Moreover,

$$\log \mathbb{P}\left(X, Y \mid z_{-u}, z_u = b\right) = \sum_{1 \leq v < w \leq n} \log f_{z_v z_w}(X_{vw}) + \sum_{v=1}^{n} \log h_{z_v}(Y_v)$$

$$= \sum_{\substack{v=1 \\ v \neq u}}^{n} \log f_{bz_v}(X_{uv}) + \log h_b(Y_u) + C$$

where $C$ does not depend on $b$. Hence

$$\Delta_{ub}(X, Y, z) = \log \frac{\pi_b}{\pi_a} + \sum_{\substack{v=1 \\ v \neq u}}^{n} \log \frac{f_{bz_v}}{f_{az_v}}(X_{uv}) + \log \frac{h_b}{h_a}(Y_u).$$

The number of nodes in block $a$ for which updating the label to $b$ would cause the change of likelihood to be strictly positive is given by

$$M_{ab}(z) = \sum_{u \in z^{-1}(a)} 1\left(\Delta_{ub}(X, Y, z) > 0\right).$$

In the following, we select $a, b$ as the indexes of the two hardest blocks to distinguish. Hence, $I = \mathrm{CH}(a, b, \pi, f, h)$ and we have using Lemma 4 that

$$\mathbb{E}_z M_{ab}(z) \sim \pi_a n e^{-(1+o(1))n\mathrm{CH}(a,b,\hat{\pi},f,g)}.$$

Since $z \in \mathcal{Z}_\xi$ we have that $\mathrm{CH}(a, b, \hat{\pi}, f, h) = (1+o(1))I$. Moreover, by assumption $I < (1-\epsilon)\frac{\log n}{n}$ for some $\epsilon > 0$ and for any $n$ large enough. Hence $\mathbb{E}_z M_{ab}(z) \gg 1$.

**(iii) Conclusion.** We will conclude that $M_{ab}(z) > 0$ almost surely using the second-moment method. Denote by $\alpha = \mathbb{P}_z(\Delta_{ub} > 0)$ and $\beta = \mathbb{P}(\Delta_{ub} > 0, \Delta_{vb} > 0)$ for two arbitrary distinct nodes $u, v \in z^{-1}(a)$. We have $\mathbb{E}_z M_{ab} = |z^{-1}(k)| \alpha$ and

$$\mathrm{Var}_z M_{ab} = \sum_{u,v \in z^{-1}(a)} \mathrm{Cov}\left(1(\Delta_{u\ell} > 0), 1(\Delta_{v\ell} > 0)\right)$$

$$= |z^{-1}(a)| \left(\alpha - \alpha^2\right) + |z^{-1}(a)| \left(|z^{-1}(a)| - 1\right) \left(\beta - \alpha^2\right)$$

$$\leq \mathbb{E}_z M_{ab} + (\mathbb{E}_z M_{ab})^2 \frac{\beta - \alpha^2}{\alpha^2}.$$

Hence, the second-moment method implies that

$$\mathbb{P}_z(M_{ab} = 0) \leq \frac{\mathrm{Var}_z M_{ab}}{(\mathbb{E}_z M_{ab})^2} \leq \frac{1}{\mathbb{E}_z M_{ab}} + \frac{\beta}{\alpha^2} - 1.$$

We end the proof using Lemma 5.

### A.4 Possibility of exact recovery

To show that the MAP estimator achieves exact recovery up to the desired threshold, we need to show that there is no possibility of reducing the likelihood by swapping $m$ vertices from each community. In all the following, we recall that $z$ denotes the true community labelling, and for any $z' \in [K]$ we denote by $L(z') = \mathbb{P}(X, Y \mid z')$ the likelihood of labelling $z'$. For any $m \geq 1$, let us denote the set of node-labelling at a distance $m$ of $z$ by

$$\Gamma_m = \{z' \in \mathcal{Z} : \mathrm{loss}(z, z') = m\}. \tag{A.8}$$

The probability that there exists a labelling $z' \in \Gamma_m$ with higher likelihood than $z$ is

$$P_m = \mathbb{P}_z\left(\exists z' \in \Gamma_m : L(z') \geq L(z)\right).$$

Let $\hat{z}$ be the block labelling estimated by the MAP. Lemma 7 ensures that $|\Gamma_m| = 0$ if $m \geq n\frac{K-1}{K}$. Thus, we have, using union bounds,

$$\mathbb{P}_z\left(\text{loss}\left(z, \hat{z}\right) \geq 1\right) \leq \sum_{m=1}^{\frac{2K-1}{2K}} |\Gamma_m| \max_{z' \in \Gamma_m} \mathbb{P}_z\left(L(z') \geq L(z)\right).$$

Moreover, [39, Proposition 5.2] show that

$$|\Gamma_m| \leq \min\left\{\left(\frac{enK}{m}\right)^m, K^n\right\},$$

while Lemma 3 provides an upper bounds on $\max_{z' \in \Gamma_m} \mathbb{P}_z\left(L(z') \geq L(z)\right)$. Combining these upper bounds ensure that for some sequence $\eta = o(1)$, we have

$$\mathbb{P}_z\left(\text{loss}(z, \hat{z}) \geq 1\right) \leq \sum_{m=1}^{n\frac{2K-1}{2K}} (s_m)^m, \tag{A.9}$$

where $s_m = \frac{enK}{m}\min\left(e^{-(1-\eta)\left(1-\frac{m}{n\pi_{\min}}\right)nI}; e^{-(n-m)d}\right)$ with $d = \sup_{t \in (0,1)} \min_{\substack{a \neq b \in [K] \\ c \in [K]}} (1 - t)\,\mathrm{D}_t(f_{ac}\|f_{bc})$.

Let $\epsilon > 0$ such that $I > (1+\epsilon)\frac{\log n}{n}$. We will now show that the sum on the right-hand side of Equation (A.9) goes to zero, by considering the cases (i) $m \leq \frac{\pi_{\min}}{2}n$ and (ii) $\frac{\pi_{\min}}{2}n \leq m \leq \frac{1}{2}n$.

(i) First of all, suppose that $m \leq \frac{\pi_{\min}}{2}n$. Then,

$$s_m \leq \frac{eK}{n^{\epsilon(1-\eta)+\eta}}e^{(1-\eta)(1+\epsilon)\frac{m}{\pi_{\min}}\frac{\log n}{n} - \log m}$$

$$\leq \frac{eK}{n^{\epsilon/2}}e^{(1+\epsilon)\frac{m}{\pi_{\min}}\frac{\log n}{n} - \log m}.$$

Let $f(m) = \frac{\log m}{m} - \frac{1+\epsilon}{\pi_{\min}}\frac{\log n}{n}$. We will show that $f(m) \geq \frac{\log m}{2m}$ for $m \geq 3$. Indeed, for $x \geq 3$, the function $x \mapsto \frac{\log x}{x}$ is decreasing. Therefore, for $3 \leq m \leq \frac{\pi_{\min}}{2}n$ we have

$$f(m) - \frac{\log m}{2m} \geq \frac{1}{2}\frac{\log m}{m} - \frac{1+\epsilon}{\pi_{\min}}\frac{\log n}{n} \geq \frac{2}{\pi_{\min}}\frac{\log n - \log(\pi_{\min}/2)}{n} - \frac{1+\epsilon}{\pi_{\min}}\frac{\log n}{n} > 0$$

for $n$ large enough. Hence $s_m \leq eKn^{-\epsilon/2}e^{-\frac{1}{2}\log m}$ and $\sum_{m=1}^{\frac{\pi_{\min}}{2}n} s_m^m = o(1)$.

(ii) Next, suppose that $\frac{1}{2}\pi_{\min}n \leq m \leq n\frac{K-1}{K}$. Then, the assumption $\mathrm{D}_t(f_{ac}\|f_{bc}) = \Theta\left(\frac{\log n}{n}\right)$ implies that $d > \kappa\frac{\log n}{n}$ for some positive constant $\kappa$. Hence,

$$s_m \leq \frac{2eK}{\pi_{\min}}e^{-\frac{\kappa}{K}\log n},$$

and thus $\sum_{m=\frac{\pi_{\min}}{2}n}^{\frac{K-1}{K}n} s_m^m = o(1)$.

Hence, (A.9) shows that $\mathbb{P}_z\left(\text{loss}(z, \hat{z}) \geq 1\right) = o(1)$, and thus the MAP estimator exactly recovers the true community structure $z$.

## A.5 Additional lemmas

**Lemma 6.** *Let $z, z' \in [K]^n$ and define for all $a, b, c, d \in [K]$:*

$$A_{ab} = |\{i \in [n]\colon (z_i, z'_i) = (a, b)\}|,$$

$$M_{(ab)(cd)} = \left|\left\{1 \leq i < j \leq n\colon (z_i, z'_i) = (a, b) \text{ and } (z_j, z'_j) = (c, d)\right\}\right|.$$

*We have*

$$M_{(ab)(cd)} = A_{ab}\left(A_{cd} - 1_{(ab)=(cd)}\right) = A_{cd}\left(A_{ab} - 1_{(ab)=(cd)}\right).$$

*Proof.* Denote by $C_a = z^{-1}(a)$ and $C'_a = z'^{-1}(a)$, and let $C_{ab} = C_a \cap C'_b$. Then, $A_{ab} = |C_{ab}|$ and $M_{(ab)(cd)} = |\{1 \le i < j \le n \colon i \in C_{ab} \text{ and } j \in C_{cd}\}|$. This proves the lemma. $\qquad\square$

**Lemma 7.** *For any $z, z' \in [K]^n$, we have $\mathrm{loss}(z, z') \le n(1 - \frac{1}{K})$.*

*Proof.* Without loss of generality, suppose that $\mathrm{loss}(z, z) = \mathrm{Ham}(z, z')$ (that is, the optimal permutation in the definition of the loss is simply the identity). For any $a \in [K]$, let us denote by $C_a = z^{-1}(a)$, $C'_a = z^{-1}(b)$. The confusion matrix of the two node labellings is the $K$-by-$K$ matrix having entries $N_{ab} = |C_a \cap C_b|$. In particular, we have $n = \sum_{a,b} N_{ab}$ and

$$
\begin{aligned}
\mathrm{loss}(z, z') &= \sum_{a=1}^{K} \sum_{\substack{b=1 \\ b \ne a}}^{K} N_{ab} \\
&= n - \sum_a N_{aa}.
\end{aligned}
$$

[5, Lemma B.2] show that $N_{ab} \le N_{aa} + N_{bb} - N_{ba}$, and therefore

$$
\mathrm{loss}(z, z') \le \sum_{a=1}^{K} \sum_{\substack{b=1 \\ b \ne a}}^{K} (N_{aa} + N_{bb} - N_{ba}). \tag{A.10}
$$

We notice that

$$
\sum_a \sum_{b \ne a} N_{aa} = (K-1) \sum_a N_{aa} = (K-1)(n - \mathrm{loss}(z, z')),
$$

and similarly,

$$
\sum_a \sum_{b \ne a} N_{bb} = \sum_a \left( \sum_b N_{bb} - N_{aa} \right) = (K-1)(n - \mathrm{loss}(z, z')).
$$

Finally, $\sum_a \sum_{b \ne a} N_{ba} = \mathrm{loss}(z, z')$. Thus, going back to (A.10) leads to

$$
\mathrm{loss}(z, z') \le 2(K-1)n - (2K-1)\mathrm{loss}(z, z'),
$$

and this ends the proof. $\qquad\square$

**Lemma 8.** *Let $f, g$ be two probability distributions and denote $c(t) = (1-t) \, \mathrm{D}_t(f\|g)$ the Chernoff coefficient of order $t$ between $f$ and $g$. We have $c'(0) = -\mathrm{d}_{\mathrm{KL}}(g\|f)$ and $c'(1) = \mathrm{d}_{\mathrm{KL}}(f\|g)$.*

*Proof.* From $c(t) = \log \int f^t g^{1-t}$, we notice that $c'(t) = \frac{\int f^t g^{1-t} \log \frac{f}{g}}{\int f^t g^{1-t}}$, and the result follows. $\qquad\square$

# B   Proof of Lemma 1

*Proof of Lemma 1.* Using a Taylor expansion, we have

$$(1-t)\,\mathrm{D}_t(f_{ac}\|f_{bc}) = \log\left[(1-p_{ac})^t(1-p_{bc})^{1-t} + p_{ac}^t p_{bc}^{1-t}\int(\tilde{f}_{ab})^t(\tilde{f}_{bc})^{1-t}\right]$$

$$= tp + (1-t)q - p^t q^{1-t}\int(\tilde{f}_{ac})^t(\tilde{f}_{bc})^{1-t} + o(\delta),$$

and we finish the proof using $\int(\tilde{f}_{ac})^t(\tilde{f}_{bc})^{1-t} = e^{-J_\psi(\theta_{ab}\|\theta_{bc})}$ and $(1-t)\,\mathrm{D}_t(h_a\|h_b) = J_\phi(\eta_a\|\eta_b)$ (see for example [28]). $\square$

# C   Additional numerical results

We present in Table 2 and 3 the numerical results that are drawn in Figure (4a) and (4b), respectively. We observe that the variance of Algorithm 1 is very low, while all other algorithms (excepted *EM-GMM*) exhibit a very large variance.

Table 2: Average ARI of the different algorithms over 60 trials with standard deviation in brackets obtained for Figure (4a).

| PARAMETER $a$ | 5 | 7 | 9 | 11 | 13 | 15 |
|---|---|---|---|---|---|---|
| ALGORITHM 1 | 0.40 (0.15) | **0.62 (0.04)** | **0.93 (0.02)** | **1 (0)** | **1 (0)** | **1 (0)** |
| EM-GMM | 0.45 (0.04) | 0.46 (0.03) | 0.46 (0.04) | 0.46 (0.04) | 0.46 (0.04) | 0.45 (0.07) |
| SC | 0 (0) | 0.07 (0.05) | 0.83 (0.15) | 0.85 (0.33) | 0.93 (0.24) | 0.82 (0.37) |
| *IR_LS* | 0 (0) | 0.20 (0.13) | 0.93 (0.14) | **1 (0)** | 0.98 (0.12) | **1 (0)** |
| *attSBM* | 0 (0) | 0.07 (0.05) | 0.82 (0.15) | 0.86 (0.33) | 0.42 (0.48) | 0.59 (0.49) |

Table 3: Average ARI of the different algorithms over 60 trials with standard deviation in brackets obtained for Figure (4b).

| PARAMETER $r$ | 0 | 1 | 2 | 3 | 4 | 5 |
|---|---|---|---|---|---|---|
| ALGORITHM 1 | 0.47 (0.29) | **0.82 (0.03)** | **0.97 (0.02)** | **1 (0)** | **1 (0)** | **1 (0)** |
| EM-GMM | 0 (0) | 0.46 (0.04) | 0.9 (0.02) | **1 (0)** | **1 (0)** | **1 (0)** |
| SC | 0.36 (0.28) | 0.38 (0.27) | 0.35 (0.26) | 0.40 (0.26) | 0.37 (0.28) | 0.35 (0.26) |
| *IR_LS* | 0.42 (0.29) | 0.65 (0.31) | 0.91 (0.21) | 0.97 (0.15) | 0.97 (0.14) | 0.98 (0.12) |
| *attSBM* | 0.36 (0.28) | 0.38 (0.27) | 0.38 (0.30) | 0.59 (0.43) | 0.57 (0.46) | 0.50 (0.43) |

We also provide in Figure 4 comparison of Algorithm 1 with *IR_sLs* and *attSBM* on weighted networks with attributes exponentially distributed. We observe that contrary to Algorithm 1, *IR_sLs* and *attSBM* do not perform well in that setting (which is expected as they are designed for binary network and Gaussian attributes).

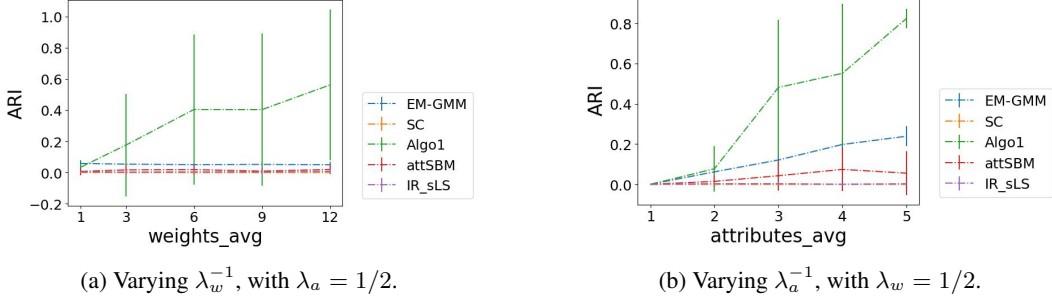

(a) Varying $\lambda_w^{-1}$, with $\lambda_a = 1/2$.    (b) Varying $\lambda_a^{-1}$, with $\lambda_w = 1/2$.

Figure 4: Performance on weighted networks ($n = 600$, $K = 2$) with edge-weight distributions $f_{\mathrm{in}} = (1-p)\delta_0 + p\,\mathrm{Exp}(\lambda_w)$, $f_{\mathrm{out}} = (1-p)\delta_0 + p\,\mathrm{Exp}(1)$ with $p = 5n^{-1}\log n$, and exponentially distributed attributes $h_1 \sim \mathrm{Exp}(1)$ and $h_2 = \mathrm{Exp}(\lambda_a)$. Results are averaged over 20 runs.

## C.1 Robustness to the choice of $d_{\psi^*}$ and $d_{\phi^*}$

We show in Figure 5 that using a divergence (distribution) for edge weights (Figure 5a and node attributes (Figure 5b different from the distribution used to generate the data does not impact the performance of Algorithm 1. We note that a similar observation was done in previous papers using Bregman divergence for clustering [6, 23].

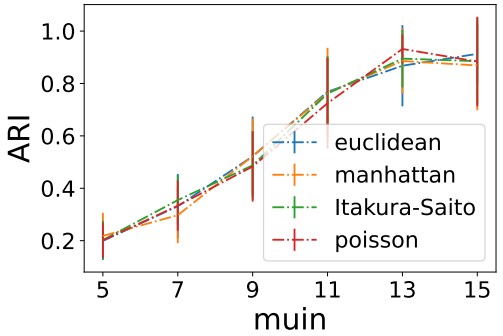
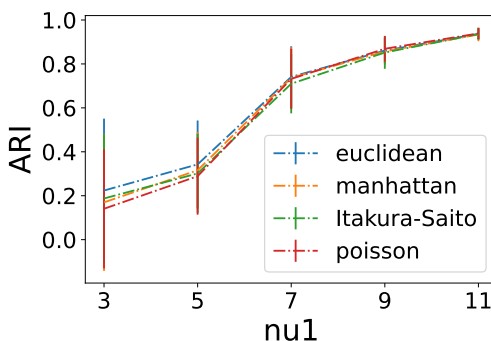

(a) Various $d_{\psi^*}$. Poisson is the correct model.  (b) Various $d_{\phi^*}$. Poisson is the correct model.

Figure 5: Performance of Algorithm 1 when $d_{\psi^*}$ or $d_{\phi^*}$ do not correspond to the model that generated the data. The different curves show the Adjusted Rand Index (ARI) [21] averaged over 20 realisations with the standard deviations as error bars.
(a) $n = 400$, $K = 4$, $f_{\text{in}} = (1-p)\delta_0(x) + p\text{Poi}(\mu_{in})$ and $f_{\text{out}} = (1-q)\delta_0(x) + q\text{Poi}(5)$, with $p = 0.04$ and $q = 0.01$. Attributes are 2d-Gaussians with unit variances and mean equally spaced the circle of radius $r = 2$.
(b) $n = 400$, $K = 2$, $f_{\text{in}} = (1-p)\delta_0(x) + p\text{Nor}(2,1)$ and $f_{\text{out}} = (1-q)\delta_0(x) + q\text{Nor}(0,1)$, with $p = 0.04$ and $q = 0.01$. Attributes are Poisson with means $\nu_1$ (for nodes in cluster 1) and 3 (for nodes in cluster 2).

