# OpenReview forum: "Exact recovery and Bregman hard clustering of node-attributed Stochastic Block Model"
_NeurIPS.cc/2023/Conference — NeurIPS 2023 poster_

### Official Review · Reviewer_1Ec2 · 2023-07-04

**Soundness:** 3 good
**Presentation:** 2 fair
**Contribution:** 2 fair
**Rating:** 6
**Confidence:** 4

**Summary:**

This paper extends the analysis of exact recovery of the Stochastic Block model to node attributed graphs (CSBM). This opens up a new dimension as the desired clustering must not only be optimal in the sense of the SBM but also respect the node attributes (which are assumed to be distributed based on the block of the SBM). First, a binary setting is presented, where edges are either present or not. For this setting, the authors present a tight bound on the information theoretical (im-)possibility of recovering the blocks of the SBM using the Chernoff information of the model. Second, a setting is analysed where the graph structure is sampled from an SBM and node attributes and edge weights are sampled from a distribution from the exponential family. For this setting, the authors present the Chernoff information of the model given its parameters allowing for a concrete characterization of the phase transition between possibility and impossibility of recovery. An iterative likelihood maximization algorithm is then proposed and evaluated on synthetic data.


**Strengths:**

- This paper provides a nice perspective on the relevance of information gained from the network structure and that gained from the node attributes.
- The claims of the paper seem sound.
- The sections themselves have clear statements.


**Weaknesses:**

- The experiments are not convincing.
	- All experiments are carried out on synthetic graphs generated in a way that benefits this algorithm.
	- The presented algorithm is compared with 2 baseline algorithms that only optimize for one of the two dimensions and thus fail at the other. Instead, one could have compared with other approaches to node-attributed community detection, which should pose more of a challenge. [19,20] could have been used as a comparison.
- The related work section is quite sparse and is used to distinguish the contribution rather than present related work. E.g. section 2.2 "Algorithms for clustering node-attributed networks" contains 2 citations. This should be augmented.
- In the presentation of the results it should be made much clearer that the developments presented here, heavily depend on [3]; with the current writing the reader gets a wrong impression of the novelty of the here presented ideas.
- The structure and the presentation of this paper are sometimes confusing.
	- The reader is often left guessing where we are headed next. E.g. section 4, which is framed as the main contribution (extending [19] by allowing for sparse networks) simply starts by introducing zero-inflated distributions from the exponential family. Or 4.2 gives the derivation for the negative log-likelihood without previously mentioning that this will be used in the iterative likelihood maximization algorithm.
	- The numbering in the appendix overwrites the numbering in the paper (e.g. Theorem 1 in the paper is not Theorem 1 in the appendix) which makes referencing the paper from the appendix or the appendix from the paper very hard.


**Questions:**

- Can you find any guarantees for your algorithm such as recovery with high probability in a certain regime?
- In the experiments, why don't you compare your approach to the other papers [19,20] that you mentioned in your related work?
- In the experiments, how close is the performance of your algorithm to the theoretical bound you presented?
- Why did you not compare the algorithms on real-world data?


**Limitations:**

Limitations of the setting or the approach are not discussed -- this should be rectified.

---

> ### Author Rebuttal · Authors · 2023-08-09
>
> Dear reviewer,
>
> thank you for the time spent reviewing our paper. Please find below answers and comments to the weaknesses and questions you have raised. We hope this will help clarify and strengthen our contribution.
>
> Weaknesses:
>
> * This is not the case. Please note that Figs. 2 and 3 of the paper compare the performance of Alg. 1 (our proposal) to 4 other algorithms. Two of them (SC and EM) only optimize for one of the dimensions (network and attributes, respectively), while the other two (attSBM and IRsls) optimize over both the network and attributes. Note that Figure 2 deals with binary networks with Gaussian weights, which is exactly the model for which both attSBM and IRsls were designed. Hence this setting is not advantageous for our Algorithm. Yet, even in this case, Alg. 1 outperforms them.
>
> Figure 3 shows that attSBM and IRsls perform terribly when the attributes are non-Gaussian and the network is non-binary, while Alg. 1 has a relatively good performance. Thus, Alg. 1 performs well in both scenarios. Moreover, we have performed experiments that show that Algorithm 1 is not too dependent on the choice of $\psi$, as questioned by reviewer coB1 (see plots in the global rebuttal file). This robustness to the choice of $\psi^*$ and $\phi^*$ confirms the superiority of Alg. 1 with respect to attSBM and IRsls.
>
> Indeed, Algorithm 1 is similar to [19] and the main (and only) difference is that their model assumes a dense weighted network (an edge is present between all node pairs with non-zero weight). As most real networks are sparse (most node pairs do not have an edge), we propose a model for sparse weighted networks (see the answer to reviewer cu9t). In any case, we have performed experiments that indicate that using a sparse model (Alg. 1) yields much better accuracy than the dense model assumed in [19, 20] when the network is sparse, as is the case for most real networks (see plots in the global rebuttal file). Moreover, since [19, 20] assume a dense network (all possible edges are present, $O(n^2)$) their algorithms do not scale to large networks. In contrast, the complexity of Alg. 1 depends on the number of edges, which is often $O(n\log n)$. We plan to revise the exposition of Section 4 and include these results and observations on complexity.
>
> * Indeed, page limitation forced us to severely restrict the discussion of the related work. The revised version will improve this discussion. Note that Section 2.2 cites the two relevant algorithms to cluster (dense) weighted networks: [9] and [20]. We mentioned [19], as it inspired Algorithm 1, but we should have described [8] and [28] as well.
>
> * The proof techniques indeed rely on [3], but also on on [33] and [5]. Yet, some arguments here are new (refer to our answer to reviewer cu9t). However, the paper clearly states that the Chernoff-Hellinger (CH) divergence was first introduced in [3] (see the introduction, section 2.1 and Remark 1). Nonetheless, as defined in [3], the expression of the CH divergence does not extend easily to non-binary interactions (contrary say, to the Renyi divergence which appears when the interaction are homogeneous SBM).
> It appears that the only work generalising the CH to a non-binary setting is [32]. Yet, in [32] the divergence is expressed as a minimisation of KL divergences, which again does not easily extend to more general interactions (say, real-valued probability distribution for edge weights). More, in [32] the link with the CH divergence requires a technical lemma (See Claim 4 of [32]), whose proof requires sparse interactions.
> Thus, in our paper expression (3.3) of the CH is novel, and not directly evident from prior works.
>
> * Indeed, the submitted paper had some typos and we appreciate your indication. A throughout reading of the paper will correct them for the revised version. We also plan to better organize the exposition of Section 4.
>
> Questions:
>
> * Unfortunately, Alg. 1 comes without any theoretical guarantees. It does not necessarily computes the MLE for the instance at hand. However, Fig. 2 in the global rebuttal file shows that Alg. 1 achieves exact recovery in a region very close to the information-theoretic limit. Indeed, two-stage algorithms have been shown to achieve optimal accuracy in binary SBMs and K-means achieve optimal accuracy in a Gaussian mixture model (see e.g., additional references [1]-[4] in comments to reviewer cu9t).
>
> * Note that the algorithms used for comparison, attSBM and IRsls in Figures 2 and 3 of the paper, are recent proposals for clustering networks with attributes (App. Net. Sci 2019 and ICML 2022, respectively). Note that algorithms proposed in [19, 20] are much older and are designed for dense networks (all possible network edges are present). In any case, we have performed experiments that indicate that using a sparse model (Alg. 1) yields much better accuracy than the dense model assumed in [19, 20] when the network is sparse, as is the case for most real networks (see plots in the global rebuttal file). Moreover, since [19, 20] assume a dense network (all possible edges are present, $O(n^2)$) their algorithms do not scale to large networks. In contrast, the complexity of Alg. 1 depends on the number of edges, which is often $O(n\log n)$. We plan to revise the exposition of Section 4 and include these results and observations on complexity.
>
> * This is a very good question. We have performed experiments with Alg. 1 comparing its exact recovery performance (fraction of times the algorithm correctly recovers the community of all nodes) with the theoretical threshold for exact recovery proved in the paper (Theorem 1). Results are shown in Fig. 2 of the global rebuttal file. Interestingly, Alg. 1 achieves exact recovery in a region very close to the information-theoretic limit. This indicates that Alg. 1 is very promising!
>
> * We have not considered real datasets. Please see comments for reviewer coB1 and hHhv.

---

> > ### Comment · Reviewer_1Ec2 · 2023-08-14
> > **response**
> >
> > I thank the authors for the response and clarifications, which have improved my understanding of the paper.
> > I thus increased my score to 6.
> >
> > I still disagree with the statement that the experiments are not favorably stacked for the algorithm considered. Indeed only 50% of the other baselines considered even consider attributes. Of those that consider attributes all assume Gaussian attributes, so it is not a surprise to see that such algorithms fail if the attributes are not Gaussian. In the case of Gaussian attributes one of the two attribute considering baselines is in fact not so much worse than what is considered here. Moreover, those baseline algorithms are designed for dense networks and not for sparse networks.
> > Ultimately, I think it would be much more convincing if the authors could represent results on real-world networks.

---

> > > ### Author Response · Authors · 2023-08-18
> > > **Preliminary results using real datasets**
> > >
> > > Thanks for the additional comment.
> > >
> > > Indeed, numerical experiments using real datasets will clearly
> > > strengthen our contribution, indicating that the proposed algorithm
> > > can also be successfully applied in the wild. Thus, we conducted
> > > preliminary experiments using the following three publicly available
> > > datasets used as benchmarks (all have no edge weights):
> > > * CiteSeer [1]: $n=3279$, $m=9104$, $K=6$, $d=3703$.
> > > * Cora [1]: $n=2708$, $m=10556$, $K=7$, $d=1433$.
> > > * Cornell [2]: $n=183$, $m=298$, $K=5$, $d=1703$.
> > >
> > > For each network, the original node attribute vector was reduced to
> > > have dimension $d=10$ by selecting the 10 best features according to
> > > the chi-square test. Since the node attribute vector in these datasets
> > > is binary, Alg. 1 assumes a Bernoulli distribution with $d=10$ for node
> > > attributes and Bernoulli edges (no edge weight). The initialization
> > > for Alg. 1 and attRBM used the spectral clustering of both the node
> > > similarity matrix (using node attributes) and network edges.
> > >
> > > Average ARI results (over independent runs) for the three datasets (in
> > > the order CiteSeer, Cora, Cornell) were as follows:
> > >
> > > * Alg. 1: 0.20, 0.12, 0.49
> > > * attRBM: 0.17, 0.09, 0.46
> > > * EM-GMM: 0.13, 0.06, 0.37
> > > * SC: 0.00, 0.00, 0.02
> > >
> > > Note that in all scenarios, Alg. 1 outperformed the other 3
> > > algorithms. Note that SC (spectral clustering of the network) has near
> > > zero performance, indicating that the network in these datasets is
> > > not informative of the clusters of the nodes. Note that both Alg. 1
> > > and attRBM (that leverage network and node attributes) outperform
> > > EM-GMM that uses only node attributes. These preliminary results
> > > indicate that Alg. 1 is promising, even when used in real datasets.
> > >
> > > Note that differently from node attributes, no preprocessing of the network
> > > was performed in the above experiments. By preprocessing the network
> > > (e.g., removing or adding edges or edge weights) it is expected that  the
> > > network can also provide information, improving the results of Alg. 1.
> > >
> > > [1] Datasets available at https://linqs.org/datasets/
> > >
> > > [2] Datasets available at https://github.com/graphdml-uiuc-jlu/geom-gcn

---

### Official Review · Reviewer_coB1 · 2023-07-04

**Soundness:** 3 good
**Presentation:** 4 excellent
**Contribution:** 3 good
**Rating:** 7
**Confidence:** 3

**Summary:**

This paper studies clustering of node-attributed Stochastic Block Model (SBM). The authors provide information-theoretic threshold for exact recovery under generic distributions for both edge weights and node attributes. In addition, the authors propose a clustering algorithm based on iterative likelihood maximization when edge weights and node attributes are drawn from exponential family distributions. The authors carry out experiment on synthetic data and show that the proposed algorithm outperforms existing state-of-the-art methods.

**Strengths:**

- The theoretical setting (i.e. clustering of node attributed SBM) considered in this work is fairly general. The threshold for exact recovery is presented in a general setting without assuming any particular family of distributions for either edge weights or node attributes. The examples and implications of Theorem 1 provided on page 5 are nice.

- The threshold which involves Equation (3.3) captures an intuitive additive signal from edge connections and node attributes.

- The overall writing and presentation is very clear and generally easy to follow (although there are many typos here and there). In particular, the presentation of background and related works seem reasonably thorough.

- The empirical results, though limited to synthetic data, are promising.

- Overall, I think this paper studies a well-known problem in a more general setting than those have been considered in prior work. The new results and algorithm can be a nice addition.



**Weaknesses:**

- Algorithm 1 requires knowing the functions $\psi^*$ and $\phi^*$. This requirement often does not hold in practice. The authors should discuss the practical implications for such requirement and what if these functions are unknown.

- Though not necessary for a theory paper, it would be nice to have at least some elementary experiments on real data.

- There are many typos throughout the paper. Here are some:
  - Line 95: standard deviation $\sigma^2$ -> standard deviation $\sigma$
  - Line 124: while his work -> while this work
  - Line 129: is find the community -> is finding the community
  - Equation (4.2): subscripts for $\theta$ do not match, similarly, subscripts for $\eta$ do not match
  - Lemma 2: $p_{k\ell}$ in the Equation should be $p_{ab}$?
  - Line 284: in Nor(...) I think the placements for mean and variance should be reversed

**Questions:**

- In practice, how should one use Algorithm 1 when $\psi^*$ and $\phi^*$ are unknown?

- Does Example 2 imply that an oracle to the true community is almost useless unless the oracle gives almost full access to all labels?

**Limitations:**

I could not find where the authors discuss the limitations or potential negative societal impact of their work.

---

> ### Author Rebuttal · Authors · 2023-08-09
>
> Dear reviewer,
>
> thank you for the time spent reviewing our paper. Please find below answers and comments to the weaknesses and questions you have raised. We hope this will help clarify and strengthen our contribution.
>
> Weaknesses:
>
> * Indeed, the distributions used to compute $\psi^*$ and $\phi^*$ are parameters of Alg. 1 and must be determined a priori, through an educated guess, for example. Note that given the distribution for edge weights and node attributed,  $\psi^*$ and $\phi^*$ are uniquely determined. Nonetheless, we have performed experiments that indicate that the performance of Alg. 1 does not strongly depend on in the right choice of $\psi^*$ (see plots in the global rebuttal file). We will add this assumption and results to the revised paper. Interestingly, a similar observation (that the right choice of distribution leads to marginal performance gains) was also made in [6] (see Tables 3 and 4 of [6]).
>
> * We have not considered real datasets because this work focuses on characterizing (theoretically) and measuring (empirically) the influence of different model parameters (e.g., edge probability, edge weights, and node attributes) on the performance of recovering the clusters. Measuring this kind of influence is quite challenging when real datasets are used. In contrast, Fig. 2 in the global rebuttal file shows the relationship between the theoretical threshold for exact recovery (theorem proved in this paper) and the performance of Alg. 1 (proposed in the paper).
>
> * Indeed, the submitted paper had some typos and we appreciate your indication. A throughout reading of the paper will correct them for the revised version.
>
> Questions:
>
> * This is a tough question and outside the scope of this paper (since we assume them to be known or guessed informatively). However, here is an idea. Since the initial clustering does not require knowledge of $\phi*$ and $\psi*$, one could treat the Bregman divergence as unknown within a set of distributions, and then learn the distribution (divergence) in the iterative steps. We will mention this as potential future work, and cite the papers [1,2] that are recent advancements in this direction.
>
> * Indeed, the exact recovery threshold is not modified unless the oracle gives almost all the labels. A high-level explanation is the following.
> Consider the planted partition model (two communities of equal size $n/2$, intra- and inter-community edge probabilities $p$ and $q$ with $p = a n^{-1} \log n$ and $q = b n^{-1} \log n)$. Exact recovery is impossible if (at least) one node has more neighbours in the opposite community than in its own community. The probability that a given node has more neighbours in the opposite community is $P_e = e^{- (1+o(1)) \frac12 ( \sqrt{a} - \sqrt{b} )^2 \log n }$ (see Section 4.1 (Lemma 4) in the last arxiv version of the review by E. Abbé, where he names this the 'genie-aided hypothesis').
> In the unsupervised setting, since there are n nodes to label, exact recovery is impossible if $n P_e$ does not go to zero. This provides the condition $(\sqrt{a} - \sqrt{b} )^2 > 2$.
> In the semi-supervised setting, there are $\eta \\, n$ nodes to cluster (where $\eta$ is the fraction of labels revealed by the oracle). If $\eta$ is constant, then the condition for exact recovery is unchanged.
> While this negative result is not new (we refer to [25]), we highlighted it here as an example since it is a special case that we find to be counter-intuitive.
>
>
> Additional references:
>
> [1] Shah, Shah, Wornell (2021). A computationally efficient method for learning exponential family distributions. Advances in neural information processing systems, 34, 15841-15854.
>
> [2] Siahkamari, Xia, Saligrama, Castañón, Kulis (2020). Learning to approximate a Bregman divergence. Advances in Neural Information Processing Systems, 33, 3603-3612.

---

### Official Review · Reviewer_cu9t · 2023-07-05

**Soundness:** 4 excellent
**Presentation:** 4 excellent
**Contribution:** 3 good
**Rating:** 6
**Confidence:** 4

**Summary:**

This paper studies community recovery in sparse, weighted networks, which is an important setting that is more general than the commonly-studied undirected, unweighted networks. The authors' first main contribution is to establish the information-theoretic conditions for exact community recovery in this setting, which is a form of the Chernoff information (and cleanly reduces to well-known info-theoretic thresholds in unweighted settings). Next, assuming that edge and node attributes belong to exponential families, an expression for the likelihood in terms of Bregman divergences is given, which then leads to a natural clustering procedure. Extensive numerical experiments are provided comparing the proposed procedure to other approaches.

**Strengths:**

Deriving the information-theoretic threshold for exact recovery via the Chernoff information is perhaps not surprising, but it is a solid contribution as it encompasses a wide range of important examples. The clustering procedure derived, based on a connection between exponential families and Bregman divergences, is clean and intuitive. The simulations on synthetic data show that the clustering procedure has favorable performance in terms of the ARI, compared to other methods, which is a good validation of the theory.

**Weaknesses:**

The paper could be improved if the authors filled in some gaps regarding a few central topics.
- Please discuss in more detail how similar / different your approach is to [19], since both leverage the connection between exponential families and Bregman divergences to come up with clustering procedures.
- The information-theoretic results are not too surprising. Does the proof require new techniques to handle the more general setting under consideration? If so, they should be discussed in the main text, as it would be of interest to theoreticians in this field.
- Please provide additional discussion around the accuracy and sample complexity of the clustering algorithm. Is it efficient? Is it guaranteed to output the MLE or similar?

**Questions:**

- You use the term "homogeneous" a lot: can you define precisely what you mean by this?
- Section 2: You can also consider citing work on Censored Block Models, in which edges take on 3 values (present, absent, unknown)
- Theorem 1: How crucial is the assumption on the convexity of CH_t (a^*, b^*) in t?
- A clarification question: if the attribute distributions were not from an exponential family, would the likelihood be intractable to compute?
- For clarity, I would suggest defining ARI.
- In the simulations, is the ARI measured between the estimated and ground-truth communities? If so, please clarify in the main text.

**Limitations:**

Yes

---

> ### Author Rebuttal · Authors · 2023-08-09
>
> Dear reviewer,
>
> thank you for the time spent reviewing our paper. Please find below answers and comments to the weaknesses and questions you have raised. We hope this will help clarify and strengthen our contribution.
>
> Weaknesses:
>
> * Indeed, Alg. 1 is similar to [19] and the main (and only) difference is that their model assumes a dense weighted network (an edge is present between any pair of nodes and has non-zero weight). As most real networks are sparse (most node pairs do not have an edge), we propose a model for sparse weighted networks. Therefore, their model assumes edge weights $w_{ij}$ are drawn from an exponential family for every node pair $(i,j)$; our model assumes edges can be present or absent (Bernoulli) and a present edge $(i,j)$ has weight $w_{ij}$ drawn from an exponential family. Note that Lemma 2 established a relationship between the log-likelihood of such zero-inflated distribution and Bregman divergences. In any case, we have performed experiments that indicate that using a sparse model (Alg. 1) yields much better accuracy than the dense model assumed in [19, 20] when the network is sparse, as is the case for most real networks (see plots in the global rebuttal file). Moreover, since [19, 20] assume a dense network (all possible edges are present, $O(n^2)$) their algorithms do not scale to large networks. In contrast, the complexity of Alg. 1 depends on the number of edges, which is often $O(n\log n)$. We plan to revise the exposition of Section 4 and include these results and observations on complexity.
>
> * The novelty in this proof is the usage of Plachky-Steinebach's theorem (a generalisation of Cramer's large deviation theorem) to derive the asymptotic behaviour of the probability that a given node $u$ is predicted to be in a wrong community by the MLE (let us call this event $Au$). Another novelty is the usage of FKG inequality to show that events $Au$ and $Av$ are positively correlated. Previous works in binary or edge-labelled SBMs typically computed this probability using ad-hoc calculations that only work for Bernoulli interactions (or interaction on a discrete, finite space). We plan to explicitly mention the Plachky-Steinebach theorem in the main text since this is essential for our theorem and may also be of interest to some readers.
>
> * Unfortunately, Alg. 1 comes without any theoretical guarantees. It does not necessarily computes the MLE for the instance at hand. However, Fig. 2 in the global rebuttal file shows that Alg. 1 achieves exact recovery in a region very close to the information-theoretic limit. Indeed, two-stage algorithms have been shown to achieve optimal accuracy in binary SBMs (see additional refs. [1]-[2]) and K-means achieve optimal accuracy in a Gaussian mixture model (additional refs. [3]-[4]). Thus, it is possible that the proposed algorithm can be analysed rigorously, but this challenging task is outside of the scope of this conference paper.
>
> Questions:
>
> * Homogeneous refers to the setting where two distributions $f$ and $g$ determine the interactions (edge weights) within and between communities, respectively. This provides for a very simple and symmetric setting. We defined it in Section 2.1, but will emphasise its meaning also in the text.
>
> * Indeed, the Censored Block Model is a nice additional example, we will add it as an example and mention recent results such as [5].
>
> * We noticed a typo in Theorem 1: the assumption should be that $CH_t(a,b)$ is strictly concave (which in turn implies the strict convexity of the function $\beta$ defined in line 71 p.4 of the Appendix). This strict convexity of $\beta$ is required to apply the Plachky-Steinebach theorem. However, this is not a constraining assumption for our setting, since the quantity $CH_t(a,b)$ is concave and will be strictly concave except in degenerate cases where all the probability distributions are equal and the divergence is zero. For example, if $f$ and $g$ are Gaussian with mean $\mu_1$, $\mu_2$ and variance 1, then $(1-t) D_t( f \| g ) = \frac12 t(1-t) (\mu_1-\mu_2)^2$ (where $D_t$ is the Renyi divergence of order $t$). This quantity is indeed strictly concave in $t$, except in the degenerate case where $\mu_1 = \mu_2$. This same argument can be made for any distribution in the exponential family. This technical comment was not mentioned in the paper due to lack of space, but we do plan to add a comment in the revised version.
>
> * Indeed, the exponential family provides some advantages when considering the MLE. For example, the Pitman-Koopman-Darmois Theorem states (under some smoothness assumptions on the probability density) that sufficient statistics with bounded dimensionality (i.e., not growing with the sample size) exist if and only if the distribution belongs to an exponential family. However, computing the MLE in weighted SBMs is computationally challenging even when restricting to exponential families, and this is the goal of Alg. 1 (not necessarily met).
>
> * Indeed, the performance of all algorithms is evaluated using the ground truth, since we have synthetic network models. We will clarify this and add a definition of the metric used (ARI).
>
> [1] Chao Gao, Zongming Ma, Anderson Y. Zhang, Harrison H. Zhou. Community detection in degree-corrected block models. The Annals of Statistics, 46(5) 2153-2185  2018.
>
> [2] Arash A. Amini, Aiyou Chen, Peter J. Bickel, Elizaveta Levina. "Pseudo-likelihood methods for community detection in large sparse networks." The Annals of Statistics, 41(4) 2097-2122 2013.
>
> [3] Yu Lu, Harrison H. Zhou (2016). Statistical and computational guarantees of Lloyd's algorithm and its variants. arXiv preprint arXiv:1612.02099.
>
> [4] Chao Gao, Anderson Y. Zhang. Iterative algorithm for discrete structure recovery. The Annals of Statistics, 50(2) 1066-1094 2022.
>
> [5] Dhara, Gaudio, Mossel, Sandon. Spectral recovery of binary censored block models. ACM-SIAM Symposium on Discrete Algorithms (SODA), 2022.

---

### Official Review · Reviewer_hHhv · 2023-07-06

**Soundness:** 4 excellent
**Presentation:** 3 good
**Contribution:** 3 good
**Rating:** 6
**Confidence:** 3

**Summary:**

This paper studies community detection in node-attributed stochastic block models. Although these models have been studied before, this work has two main contributions: (i) the edge weights in the model are now not necessarily binary, but can also be weighted. The node attributes don’t have to be Gaussian, and can come from a more general exponential family. (ii) An iterative clustering algorithm which maximizes the likelihood by placing nodes in correct cluster based on attributes and edge weights.
Experiments on synthetic datasets are also presented


**Strengths:**

•	The weakening of the assumptions in this work are more general than what previous work has considered. This is a significant strength of this work. The assumptions in this model (weighted edges, non Gaussian node attributes) are more realistic than what previous work has considered, and I consider this a nice contribution.

•	To the best of my knowledge the theoretical work is sound and contains some non-trivial insights. Some of the technical techniques presented might have impact on future work.

•	Experimental results on synthetic data suggest that the proposed algorithm 1 performs well compared to previous work.



**Weaknesses:**

•	Given that a motivation for this paper is to provide an algorithm for a more realistic setting, I would have expected to see experiments on real world datasets. In general, stochastic block models enjoy a very symmetric structure, which in some sense makes it nice to cluster. It would be interesting to see how the performance of this algorithm generalises to real-world graphs.

•	Experimental results on synthetic data is only performed when $k=2$. Higher values of $k$ should be considered


## Minor:

Line 73: conditioned of --> conditioned on

Line 129: is find --> is to find


**Questions:**

None

---

> ### Author Rebuttal · Authors · 2023-08-09
>
> Dear reviewer,
>
> Thank you for the time spent reviewing our paper. In order to address the weaknesses pointed out in your review, we have performed novel experiments using synthetic data sets with a larger number of clusters ($k=3$ and $k=4$). Results using more clusters are qualitatively similar to the $k=2$ case but do strengthen the results in the paper (see plots in the global rebuttal file).
>
> We have not considered real datasets because this work focuses on characterizing (theoretically) and measuring (empirically) the influence of different model parameters (e.g., edge probability, edge weights, and node attributes) on the performance of recovering the clusters. Measuring this kind of influence is quite challenging when real datasets are used. In contrast, Fig. 2 in the global rebuttal file shows the relationship between the theoretical threshold for exact recovery (theorem proved in this paper) and the performance of Alg. 1 (proposed in the paper).

---

> > ### Comment · Reviewer_hHhv · 2023-08-11
> > **Acknowledgement Response**
> >
> > I thank the authors for their additional experiments on larger $k$ which do improve the experimental results in this work.
> >
> > I do still think that real-world experiments (albeit in a slightly adjusted form) would significantly strengthen the paper. I do appreciate the theoretical analysis of this work which is non-trivial and important (hence my positive evaluation of the paper). However, a large part of the introduction is spent justifying the additional parameters (for continuous edge weights) in the CSBM because of real-world observations. One would therefore also expect some form of real-world experiments.

---

> > > ### Author Response · Authors · 2023-08-18
> > > **Preliminary results using real datasets**
> > >
> > > Thanks for the additional comment.
> > >
> > > Indeed, numerical experiments using real datasets will clearly
> > > strengthen our contribution, indicating that the proposed algorithm
> > > can also be successfully applied in the wild. Thus, we conducted
> > > preliminary experiments using the following three publicly available
> > > datasets used as benchmarks (all have no edge weights):
> > > * CiteSeer [1]: $n=3279$, $m=9104$, $K=6$, $d=3703$.
> > > * Cora [1]: $n=2708$, $m=10556$, $K=7$, $d=1433$.
> > > * Cornell [2]: $n=183$, $m=298$, $K=5$, $d=1703$.
> > >
> > > For each network, the original node attribute vector was reduced to
> > > have dimension $d=10$ by selecting the 10 best features according to
> > > the chi-square test. Since the node attribute vector in these datasets
> > > is binary, Alg. 1 assumes a Bernoulli distribution with $d=10$ for node
> > > attributes and Bernoulli edges (no edge weight). The initialization
> > > for Alg. 1 and attRBM used the spectral clustering of both the node
> > > similarity matrix (using node attributes) and network edges.
> > >
> > > Average ARI results (over independent runs) for the three datasets (in
> > > the order CiteSeer, Cora, Cornell) were as follows:
> > >
> > > * Alg. 1: 0.20, 0.12, 0.49
> > > * attRBM: 0.17, 0.09, 0.46
> > > * EM-GMM: 0.13, 0.06, 0.37
> > > * SC: 0.00, 0.00, 0.02
> > >
> > > Note that in all scenarios, Alg. 1 outperformed the other 3
> > > algorithms. Note that SC (spectral clustering of the network) has near
> > > zero performance, indicating that the network in these datasets is
> > > not informative of the clusters of the nodes. Note that both Alg. 1
> > > and attRBM (that leverage network and node attributes) outperform
> > > EM-GMM that uses only node attributes. These preliminary results
> > > indicate that Alg. 1 is promising, even when used in real datasets.
> > >
> > > Note that differently from node attributes, no preprocessing of the network
> > > was performed in the above experiments. By preprocessing the network
> > > (e.g., removing or adding edges or edge weights) it is expected that  the
> > > network can also provide information, improving the results of Alg. 1.
> > >
> > > [1] Datasets available at https://linqs.org/datasets/
> > >
> > > [2] Datasets available at https://github.com/graphdml-uiuc-jlu/geom-gcn

---

### Author Rebuttal · Authors · 2023-08-09

First and foremost, we would like to thank all four reviewers for their time spend reviewing our paper and their valuable comments.
Some of the questions raised by the reviewers can be addressed with further experiments, as the following:

* how does Algorithm 1 perform when the network has more clusters?
* how sensitive is the performance of Algorithm 1 to the choice of the divergences $d_{\psi^*}$ and $d_{\phi^*}$?
* how does the performance of Algorithm 1 compare to the theoretical threshold for exact recovery derived in Section 3?
* how does Algorithm 1 compare to algorithms published in [19] and [20]?


The attached PDF file has figures with the results of these questions. More precisely:


* Figures 1(a) and 3(c) have 4 clusters, and Figure 2(b) has 3 clusters. (Note that all these new experiments were run with same size clusters, and respective intra-cluster and inter-cluster edge densities $f_{in}$ and $f_{out}$)

* Figure 1 shows that using a divergence (distribution) for edge weights (Fig. 1(a)) and node attributes (Fig. 1(b)) different from the distribution used to generate the data does not impact the results.

* Figure 2 compares the performance of Alg. 1 in terms of exact recovery (fraction of times the algorithm correctly recovers the community of all nodes) with the theoretical threshold for exact recovery proved in the paper (red curve in the plots) in two settings:
(2a) binary weight with Gaussian attributes, and (2b) zero-inflated Gaussian weights with Gaussian attributes. Solid black and white squares represent fraction zero (no trial was recovered exactly) and one (all trials were exactly recovered) over 50 trials.

This is a very important numerical validation of Algorithm 1, and we plan to include it in the paper (to replace Figure 1 of the article, which does not explicitly show the theoretical curve). Due to limited time in the rebuttal phase, this new Figure 2 has a relatively large granularity and each pixel is averaged over 50 runs, but we can increase these numbers for the final version.

* Figures 3(a) and (b) compare Algorithm 1 to the V-EM algorithm of [20] and to the Bregman algorithm of [19]. Note that algorithm [19] is a special case of Algorithm 1 for when the network is dense (all possible edges are present). Results indicate that when the network is sparse, [19] has poor performance (and not surprising since it assumes a dense network). The V-EM of [20] is also designed for dense networks, and while its performance on sparse networks is not poor, Algorithm 1 is superior (and not surprising, since it was designed for sparse networks). Last, since the models in [19, 20] assume a dense network (all possible edges are present, $O(n^2)$) their algorithms do not scale to large networks. In contrast, the complexity of Alg. 1 depends on the number of edges, which is often $O(n\log n)$. (The three algorithms had the same initialisation, and the performance of the initialization is given as a reference of a baseline to beat).

* Finally, Figure 3(c) is similar to Figure 2(b) of the paper (binary network with Gaussian attributes) but with a larger number of clusters. We see that Algorithm 1 performs better than the algorithm of [8], which was a recently proposed algorithm to tackle sparse networks with binary edges and Gaussian attributes.

References:

[6] Arindam Banerjee, Srujana Merugu, Inderjit S Dhillon, Joydeep Ghosh, and John Lafferty. Clustering with Bregman divergences. Journal of machine learning research, 6(10), 2005.

[8] Guillaume Braun, Hemant Tyagi, and Christophe Biernacki. An iterative clustering algorithm for the contextual stochastic block model with optimality guarantees. In International Conference on Machine Learning, pages 2257–2291. PMLR, 2022.

[19] Bo Long, Zhongfei Mark Zhang, and Philip S Yu. A probabilistic framework for relational clustering. In ACM International Conference on Knowledge Discovery and Data Mining (SIGKDD), pages 470–479, 2007.

[20] Mahendra Mariadassou, Stéphane Robin, and Corinne Vacher. Uncovering latent structure in valued graphs: A variational approach. The Annals of Applied Statistics, 4(2):715 – 742, 2010.

---

### Decision · Program_Chairs · 2023-09-21

**Decision:**

Accept (poster)

**Comment:**

All reviewers agree to accept this paper. The paper is theoretically sound and sufficiently novel to grant acceptance at this conference. Experiments on synthetic data complement the theoretical results.

For a camera-ready version, please take into the account the comments from all reviewers regarding typos and clarifications. In particular, please include:
- a discussion of the novel techniques used in the proofs
- synthetic experiments for $k=3$ and $k=4$ clusters
- real-world experiments on CiteSeer, Cora, Cornell
- a discussion and experimental comparison with [19] and [20]